# A naturalistic environment to study visual cognition in unrestrained monkeys

Georgin Jacob[1,2†], Harish Katti[1†], Thomas Cherian[1‡], Jhilik Das[1‡], KA Zhivago[1], SP Arun[1]*

[1]Centre for Neuroscience, Indian Institute of Science Bangalore, Bangalore, India; [2]Department of Electrical Communication Engineering Indian Institute of Science, Bangalore, India

**Abstract** Macaque monkeys are widely used to study vision. In the traditional approach, monkeys are brought into a lab to perform visual tasks while they are restrained to obtain stable eye tracking and neural recordings. Here, we describe a novel environment to study visual cognition in a more natural setting as well as other natural and social behaviors. We designed a naturalistic environment with an integrated touchscreen workstation that enables high-quality eye tracking in unrestrained monkeys. We used this environment to train monkeys on a challenging same-different task. We also show that this environment can reveal interesting novel social behaviors. As proof of concept, we show that two naive monkeys were able to learn this complex task through a combination of socially observing trained monkeys and solo trial-and-error. We propose that such naturalistic environments can be used to rigorously study visual cognition as well as other natural and social behaviors in freely moving monkeys.

*For correspondence: sparun@iisc.ac.in

†These authors contributed equally to this work
‡These authors also contributed equally to this work

Competing interest: The authors declare that no competing interests exist.

## Editor's evaluation

The manuscript describes a naturalistic experimental environment for training and testing macaque monkeys and for recording head-unrestrained eye movements. The utility of the setup is demonstrated through eye movement and social learning data during a cognitive (same-different) task. The authors conclude that this new environment provides a promising platform for studying cognitive and social behaviors, potentially in conjunction with wireless neurophysiological recordings in the future.

## Introduction

Macaque monkeys are highly intelligent and social animals with many similarities to humans, due to which they are widely used to understand cognition and its neural basis (*Passingham, 2009*; *Roelfsema and Treue, 2014*; *Buffalo et al., 2019*). In the traditional approach for studying vision, monkeys are brought into a specialized lab where the head is restrained to obtain non-invasive eye tracking and minimize movement artifacts during neural recordings. This approach prevents a deeper understanding of vision in more natural, unrestrained settings.

However, studying vision in a more natural setting requires overcoming two major challenges. First, animals must be housed in a naturalistic environment to engage in natural, social behaviors while at the same time repeatedly access complex cognitive tasks as required for the rigorous study of behavior and cognition. The design principles for such naturalistic environments as well as standard procedures to maximize animal welfare are well understood now (*Woolverton et al., 1989*; *Röder and Timmermans, 2002*; *Honess and Marin, 2006*; *Seier et al., 2011*; *Cannon et al., 2016*; *Coleman and Novak, 2017*). Recent studies have demonstrated that monkeys can be trained to

perform complex tasks using touchscreen devices that can be easily integrated into a naturalistic environment (*Rumbaugh et al., 1989*; *Mandell and Sackett, 2008*; *Fagot and Paleressompoulle, 2009*; *Gazes et al., 2013*; *Calapai et al., 2017*; *Claidière et al., 2017*; *Tulip et al., 2017*; *Berger et al., 2018*). While there are rigorous approaches to evaluate group performance on various tasks (*Drea, 2006*), it should also be possible to separate individual animals from the group to assess their individual performance on complex tasks.

Second, it should be possible to obtain high-fidelity gaze tracking in unrestrained macaque monkeys. All commercial eye trackers work best when the head is in a stereotypical front-facing position with relatively little movement, and their gaze tracking degrades with any head movement. As a result, obtaining accurate gaze signals from unrestrained animals can be a major challenge (for a review of existing literature and best practices, see *Hopper et al., 2021*). Most studies of macaque eye tracking require some form of head restraint while monkeys are seated in a monkey chair (*Machado and Nelson, 2011*; *De Luna and Rainer, 2014*; *Kawaguchi et al., 2019*; *Ryan et al., 2019*). Another solution is to use wearable eye trackers, but these require extensive animal training to avoid equipment damage (*Milton et al., 2020*). A further complication is that most eye trackers are optimized for larger screen distances (~60 cm) which allow for shallow angles between the eye tracker line-of-sight and the screen (*Hopper et al., 2021*). By contrast, a macaque monkey reaching for a touchscreen requires far smaller distances (~20 cm), resulting in elevated angles for the eye tracker, all of which compromise tracking quality. Finally, many commercial eye-tracking systems are optimized for the human inter-pupillary distance (~60 mm) as opposed to that of monkeys (~30 mm), which also result in compromised gaze tracking ability.

Here, we designed a naturalistic environment with a touchscreen workstation and an eye tracker to study natural behaviors as well as controlled cognitive tasks in freely moving monkeys. We demonstrate several novel technical advances: (1) We show that, even though the monkeys can freely move to approach or withdraw from the workstation, their gaze can be tracked in real-time with high fidelity whenever they interact with the touchscreen for juice reward. This was possible due to a custom-designed juice spout with a chin-rest that brought the monkey into a stereotyped head position every time it drank juice, and by adjusting the eye tracker illuminator and camera positions; (2) We show that this enables gaze-contingent tasks and high-fidelity eye tracking, both of which are crucial requirements for studying visual cognition. (3) We show that this environment can be used to train monkeys on a complex same-different task by taking them through a sequence of subtasks of increasing complexity. (4) Finally, we illustrate how this novel environment can reveal interesting behaviors that would not have been observable in the traditional paradigm. Specifically, we show that naive monkeys can rapidly learn a complex task through a combination of socially observing trained monkeys perform the task at close quarters, and through solo sessions with trial-and-error learning. These technical advances constitute an important first step toward studying vision in a more natural setting in unrestrained, freely moving monkeys.

## Results

### Environment overview

We designed a novel naturalistic environment for studying cognition during controlled cognitive tasks as well as natural and social behaviors (*Figure 1*). Monkeys were group-housed in an enriched living environment with access to a touchscreen workstation where they could perform cognitive tasks for juice reward (*Figure 1A*; see Materials and methods). The enriched environment comprised log perches and dead trees with natural as well as artificial lighting with several CCTV cameras to monitor movements (*Figure 1B*). We also included tall perches for animals to retreat to safety (*Figure 1C*). The continuous camera recordings enabled us to reconstruct activity maps of the animals with and without human interactions (*Figure 1D*; *Video 1*). To allow specific animals access to the behavior room, we designed a corridor with movable partitions so that the selected animal could be induced to enter while restricting others (*Figure 1*). We included a squeeze partition that was not used for training but was used if required for administering drugs or for routine blood testing (*Figure 1F*). This squeeze partition had a ratchet mechanism and locks for easy operation (*Figure 1G*). After traversing the corridor (*Figure 1H*), monkeys entered a behavior room containing a touchscreen workstation (*Figure 1I*). The behavior room contained copper-sandwiched high pressure laminated panels that formed a closed

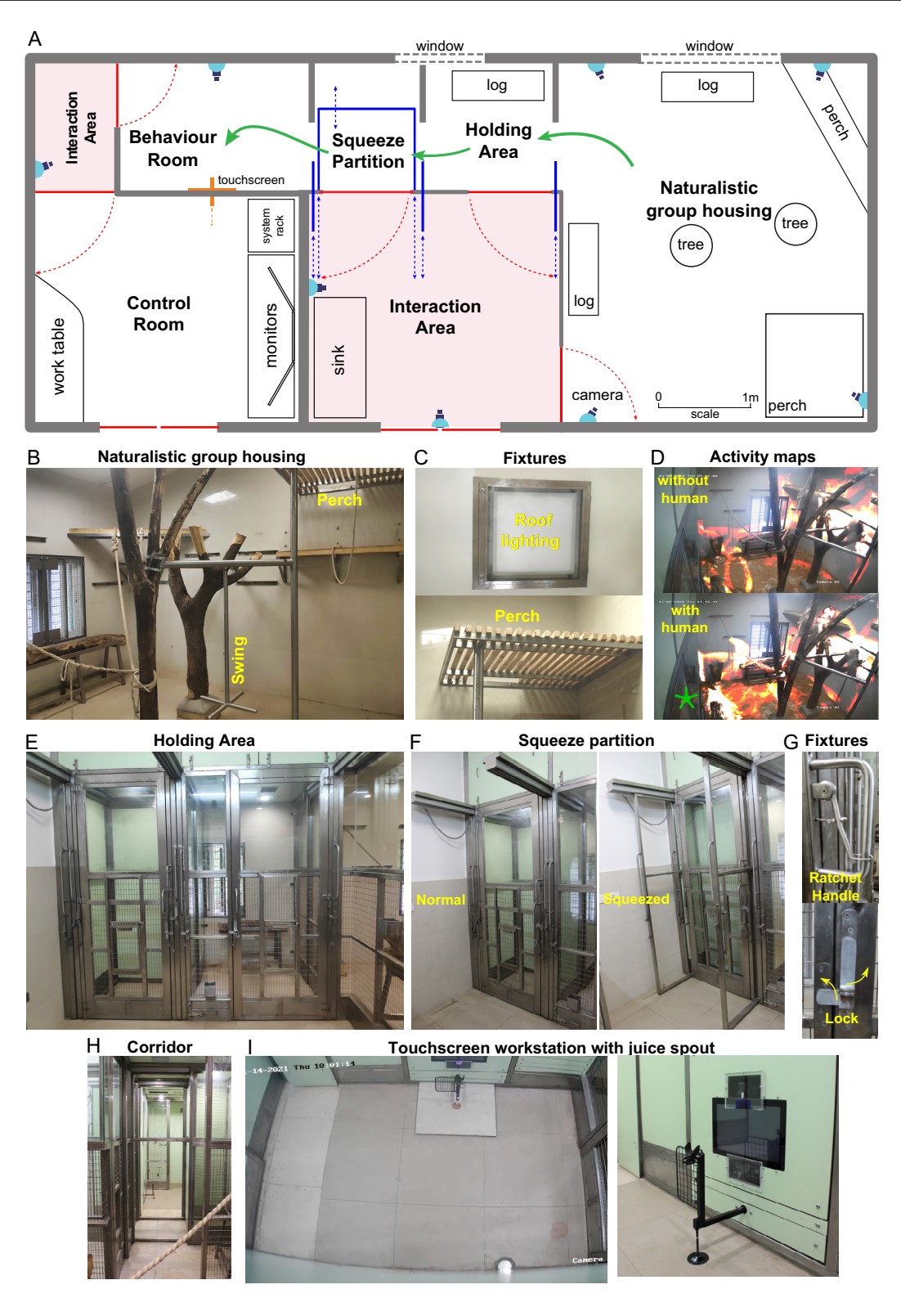

**Figure 1.** Overview of naturalistic environment. (**A**) Illustrated layout of the environment designed to enable easy access for monkeys to behavioral tasks. Major features placed for enrichment are labelled. *Blue lines* indicate partitions for providing access to various portions of the play area. Typical movement of an animal is indicated using *green arrows. Red lines* indicate doors that are normally kept closed. (**B**) View into the play area from the interaction room showing the enriched environment. (**C**) *Top:* Roof lights that have been enclosed in stainless steel and toughened glass case to be

*Figure 1 continued on next page*

*Figure 1 continued*

tamper-proof. *Bottom:* Close up of the perch that provides monkeys with an elevated point of observation. (**D**) *Top:* Heatmap of residence duration of monkeys (red to yellow to white = less to more time spent in location) in the play area analyzed from a ~ 7 min video feed of one of the CCTV cameras. There was no human presence in the interaction room during this period. *Bottom:* The same residence analysis but with human presence in the interaction room during a ~ 7 min period on the same day. See *Video 1* (**E**) View from below the CCTV in the interaction area onto the squeeze and holding areas with trap-doors affixed to bring the monkey out into a chair when required. (**F**) The squeeze partition for temporarily restraining monkeys . *Left:* View of the partition in the normal open condition *Right:* View of the partition in the squeezed condition. (**G**) *Top:* Close-up view of the rachet mechanism to bring the squeeze partition forward. *Bottom:* Close-up view of the monkey-proof lock on each door. (**H**) View of the path taken by monkeys from play area through the holding and squeeze area into the behavior room. (**I**) *Left:* Top-down view from the CCTV in the behavior room showing the placement of the touchscreen on the modular panel wall and the juice reward arm in front of it. *Right:* Close-up view of the touchscreen and the juice reward arm.

circuit for removing external electromagnetic noise, to facilitate eventual brain recordings (*Figure 2—figure supplement 2*). The entire workflow was designed so that experimenters would never have to directly handle or contact the animals during training. Even though the environment contained safe perches out of reach from humans, we were able to develop standard protocols to isolate each monkey and give it access to the behaviour room (see Materials and methods).

## Touchscreen workstation with eye tracking in unrestrained monkeys

The touchscreen workstation is detailed in *Figure 2*. Monkeys were trained to sit comfortably at the juice spout and perform tasks on the touchscreen for juice reward. The workstation contained several critical design elements that enabled behavioral control and high-fidelity eye tracking, as summarized below (see *Video 2*).

First, we developed a juice delivery arm with a drain pipe that would take any extra juice back out to a juice reservoir (*Figure 2—figure supplement 3*). This was done to ensure that monkeys drank juice directly from the juice spout after a correct trial instead of subverting it and accessing spillover juice. Second, we developed several modular head frames that were tailored to the typical shape of the monkey head (*Figure 2B*; *Figure 2—figure supplement 3*). In practice, monkeys comfortably rested their chin/head on these frames and were willing to perform hundreds of trials even while using the most restrictive frames. Third, we affixed two transparent viewports above and below the touchscreen, one for the eye tracker camera and the other for the infrared radiation (IR) illuminator of the eye tracker respectively (*Figure 2A–B*). Finally, we included a removable hand grill to prevent the monkeys from accessing the touchscreen with the left hand (*Figure 2A*). This was critical not only for reducing movement variability but also to provide an uninterrupted path for the light from the IR illuminator of the eye tracker mounted below the touchscreen to reflect off the eyes and reach the eye tracker camera mounted above the touch screen (*Figure 2A–B*). This design essentially stereotyped the position of the monkey's head and gave us excellent pupil and eye images (*Figure 2C*, inset) and consequently highly accurate eye tracking (see *Video 2*).

## Same-different task with gaze-contingent eye tracking

Understanding visual cognition often requires training monkeys on complex cognitive tasks with events contingent on their eye movements, such as requiring them to fixate or make saccades. As a proof of concept, we trained two animals (M1 & M3) on a same-different (i.e., delayed match-to-sample) task with real-time gaze-contingency.

The timeline of the task is depicted schematically in *Figure 3A*. Each trial began with a hold cue that was displayed until the animal touched it with his hand, after which a fixation cross appeared at the center of the screen. The monkey had to keep its hand on the hold cue and maintain its gaze within a 8° radius around the fixation cross. Following this a sample image appeared for 500 ms after which the screen went blank for

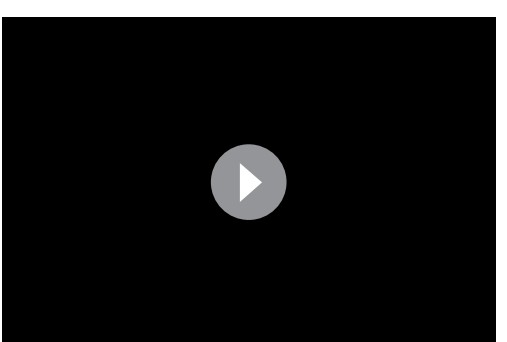

**Video 1.** Monkey movement in play area.
https://elifesciences.org/articles/63816/figures#video1

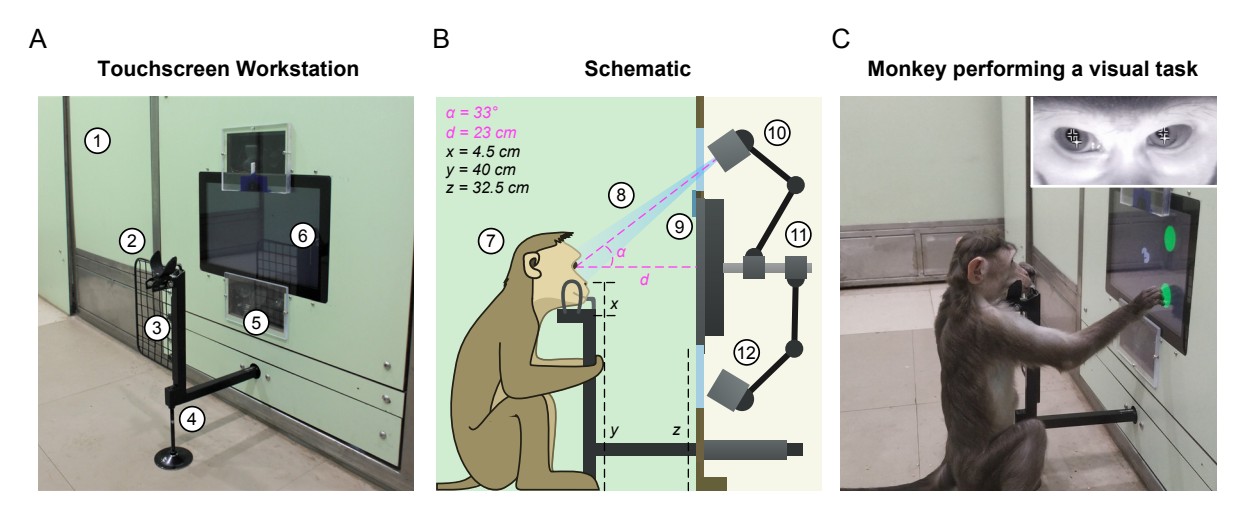

**Figure 2.** Touchscreen workstation with eye tracking for unrestrained monkeys. (**A**) Labeled photograph of the touchscreen workstation from the monkey's side. *Labels: 1: Partition panel with electromagnetic shielding; 2: Chin rest; 3: Grill to block left-hand screen access; 4: Movable reward delivery arm with concealed juice pipe; 5: Transparent viewports 6: Touchscreen.* (**B**) Labeled cross-section showing both monkey and experimenter sides. *Labels: 7: Position of monkey at the workstation; 8: Field of view of the eye tracker; 9: Channel for mounting photodiode; 10: Eye tracker camera and additional synchronized optical video camera; 11: Adjustable arms mounted on the shaft behind touchscreen back panel; 12: Eye tracker IR illuminator.* (**C**) Photograph of monkey M1 performing a task. . *Inset: Screengrab from the ISCAN IR eye tracker camera feed while monkey was doing the task, showing the detected pupil (black crosshair with white border) and corneal reflection (white crosshair with black border).*

The online version of this article includes the following figure supplement(s) for figure 2:

**Figure supplement 1.** System components and technical specifications.

**Figure supplement 2.** Electromagnetic shielding and reward system.

**Figure supplement 3.** Custom juice spout and snout restraints.

200 ms. After this, several events happened simultaneously: a test stimulus appeared, the hold cue disappeared, fixation/hold constraints were removed, and two choice buttons appeared above and below the hold cue. The animal had to make a response by touching one of the choice buttons within 5 s. The test stimulus and the choice buttons were presented till the monkey made a response, or till 5 s, whichever is earlier. If the test image was identical to the sample, the monkey had to touch the upper button or if it was different, the lower button. Example videos of the same-different task and a more complex part-matching task are shown in *Video 3*.

*Figure 3B* illustrates the example gaze data recorded from monkey M1 during two trials of the same-different task, one with a 'SAME' response and the other with a 'DIFFERENT' response. The monkey initially looked at the hold button, then at the sample image, and eventually at the choice buttons. The time course of the two trials reveals eye movements in the expected directions: for the 'SAME' trial, the vertical eye position moves up shortly after the test stimulus appeared (*Figure 3C*), whereas in a 'DIFFERENT' trial, the vertical position moves down (*Figure 3D*). We obtained highly reliable gaze position across trials (*Figure 3E*), allowing us to reconstruct the characteristic time course of saccades (*Figure 3F–G*). We obtained similar, highly reliable gaze signals from another animal M3 as well (*Figure 3—figure supplement 1*). This accuracy is remarkable given that this is from entirely unrestrained monkeys.

To characterize the quality of fixation in this setup, we analyzed the gaze data across many hundreds of trials for monkey M1. By comparing our networked video cameras with the eye tracker gaze position signals, we found that gaze data

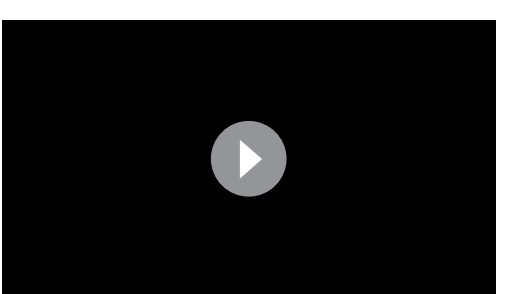

**Video 2.** Eye tracking during a same-different task.
https://elifesciences.org/articles/63816/figures#video2

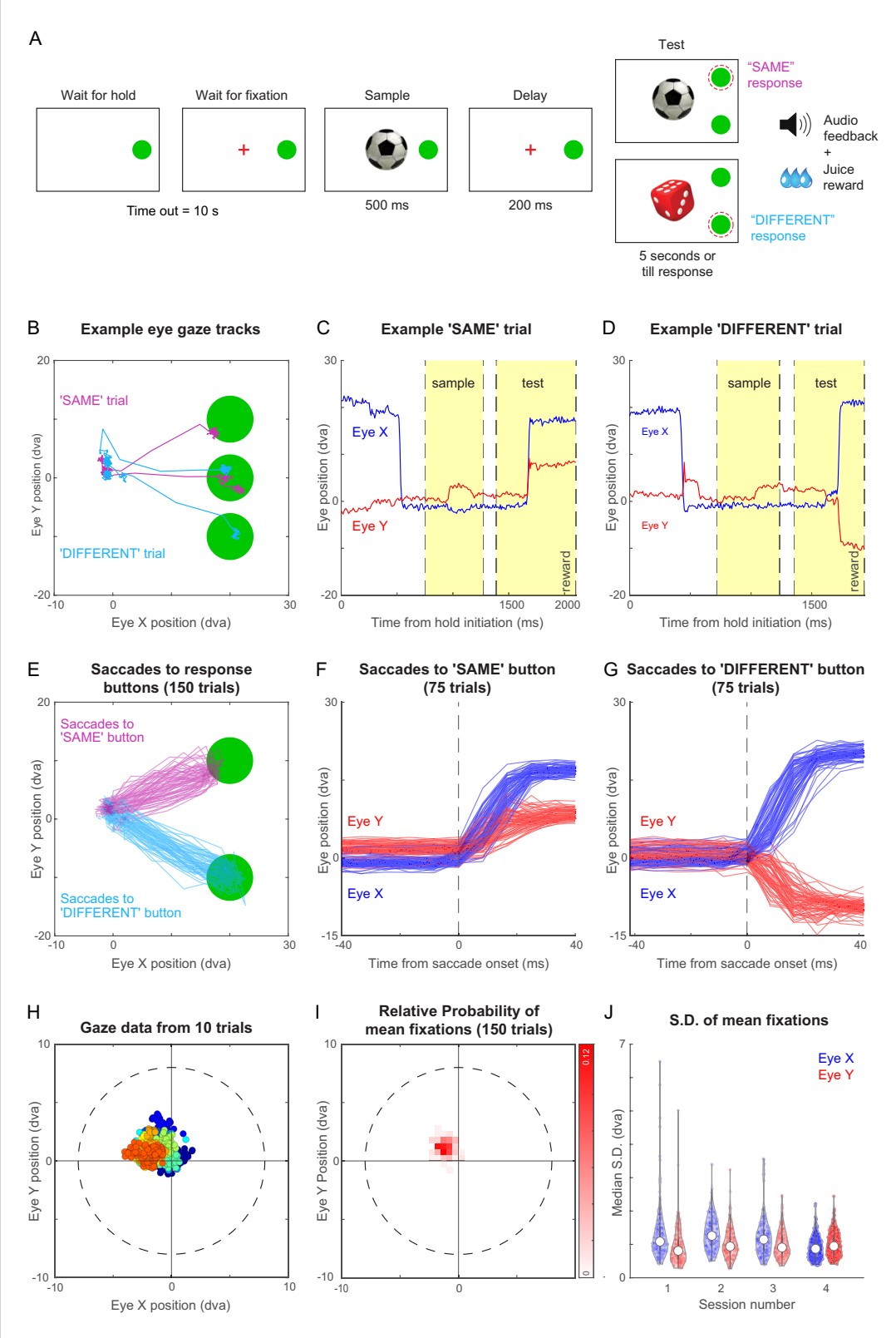

**Figure 3.** Same-different task with gaze-contingent tracking for monkey M1. (**A**) Schematic sequence of events in the same-different task. The monkey had to touch the HOLD button and look at a fixation cross at the centre of the screen, after which a sample stimulus appeared for 500 ms followed by a blank screen for 200 ms. Following this a test stimulus appeared along with choice buttons for SAME and DIFFERENT responses. The monkey had to indicate by touching the appropriate button whether the sample and test were same or different. All trials were followed by different audio tones for

*Figure 3 continued on next page*

*Figure 3 continued*

correct and error trials, and the monkey received juice for correct trials. See *Video 2* . (**B**) Eye traces overlaid on the stimulus screen, for one example SAME response trial (*magenta*) and one representative DIFFERENT trial (*cyan*) for monkey M1. (**C**) Horizontal (*blue*) and vertical (*red*) gaze position as a function of time during the SAME trial shown in (**A**). Dotted lines mark sample on, sample off, test on, and reward (from left to right respectively, along the x-axis). (**D**) Same as (**C**) but during a correct DIFFERENT choice trial in (**A**). (**E**) Horizontal and vertical gaze position during SAME response trials (*magenta*) and DIFFERENT response trials (*cyan*) over a total of 150 trials (75 SAME trials and 75 DIFFERENT trials). (**F**) Gaze position as a function of time (aligned to saccade onset) for the SAME response trials shown in (**E**). Saccade onset was defined based on the time at which saccade velocity attained 10% of the maximum eye velocity. (**G**) Same as (**F**) but for DIFFERENT response trials. (**H**) Gaze positions during 10 example trials during the fixation-contingent period in Session 4. The monkey had to maintain gaze during this period within a fixation window of 8 dva radius (dotted circle) centred at the middle of the screen (where sample and fixation spot were presented). Data from individual trials are shown in different colours. (**I**) 2D histogram of the mean gaze position in each trial across all 150 trials in (**E**) from Session 4. (**J**) Violin plot showing the standard deviation of gaze positions within each trial for both horizontal (Eye X) and vertical (Eye Y) directions across trials in four separate sessions (Sessions 1–4, where session four data is the same in panels B to I), overlaid with median (*white dot*) and inter-quartile range (*vertical gray bar*).

The online version of this article includes the following figure supplement(s) for figure 3:

**Figure supplement 1.** Eye tracking during same-different task for monkey M3.

**Figure supplement 2.** Eye tracking during a fixation task for Monkeys M1 & M3.

was missing if and only if the animal looked away or moved away from the touchscreen, with no gaze data lost when the monkeys did not look away. Although we imposed a relatively liberal fixation window (radius = 8°), the animals' eye positions were far more concentrated within a given trial with average gaze position changing slightly from trial to trial (*Figure 3H*). To quantify these patterns, we plotted the distribution of average gaze position across 150 trials for monkey M1 (*Figure 3I*). It can be seen that the center of gaze was slightly northwest of the center estimated by the gaze calibration. To quantify the fixation quality within each trial, we calculated the standard deviation along horizontal and vertical directions for each trial. This revealed gaze to be tightly centered with a small standard deviation (standard deviation, mean ± s.d. across 150 trials: 0.90° ± 0.36° along x, 1.01° ± 0.38° along y). We obtained similar, tightly centered standard deviation across sessions (*Figure 3J*). We obtained qualitatively similar results for monkey M3 in the same-different task. (*Figure 3—figure supplement 1*). Interestingly, the eye tracking revealed that monkey M3 looked first at the DIFFERENT button by default and then made a corrective saccade to the SAME button (*Figure 3—figure supplement 1*). Finally, we also trained both monkeys M1 and M3 on a fixation task and obtained highly accurate eye tracking and fixation quality in both monkeys (*Figure 3—figure supplement 2*).

This high fidelity of gaze data in unrestrained monkeys was due to two crucial innovations. First, the stereotyped position of the juice spout made the animal put its head in exactly the same position each time, enabling accurate eye tracking (*Video 2*). Second, the eye tracker camera and IR illuminator were split and placed above and below the screen, enabling high-quality pupil and corneal reflections, boosting tracking fidelity.

## Tailored automated training (TAT) on same-different task

Here we describe our novel approach to training animals on this same-different task, which we term as 'Tailored Automated Training' (TAT). In the traditional paradigm, before any task training can be started, monkeys have to be gradually acclimatized to entering specialized monkey chairs that block them from access to their head, and to having their head immobilized using headposts for the purpose of eye tracking. This process can take a few months and therefore is a major bottleneck in training (*Fernström et al., 2009*; *Slater et al., 2016*; *Mason et al., 2019*). These steps are no longer required in our environment, allowing us to focus entirely on task-relevant training.

We trained two monkeys (M1 and M3) using TAT (for details, see Appendix 1). The fundamental approach to training monkeys on complex tasks is to take the animal through several stages

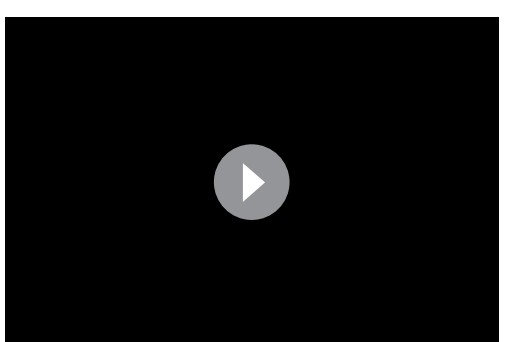

**Video 3.** Same-different task variations.
https://elifesciences.org/articles/63816/figures#video3

of gradual training so that at every stage the animal is performing above chance, while at the same time learning continuously. On each session, we gave access to the touchscreen workstation to each monkey individually by separating it from its group using the holding areas (*Figure 1A*). Each monkey was guided automatically through increasingly complex stages of the same-different task. These stages went from a basic task where the monkey received a reward for touching/holding a target square on the screen, to the full same-different task described in the previous sections. Importantly, each monkey went through a unique trajectory of learning that was tailored to its competence on each stage. There were a total of 10 stages and multiple levels within each stage. Only one task-related parameter was varied across levels in any given stage. The monkey would progress to the next level once it completed most recent 50 trials with at least 80% accuracy. By the end of training, both monkeys were highly accurate on the same-different task (91% for M1, 82% for M3). The duration of training from completely naïve to fully trained was approximately 90 sessions or days. Thus, the tailored automated training (TAT) paradigm deployed in this naturalistic environment can enable automated training of monkeys on complex cognitive tasks while at the same time maximizing animal welfare.

## Can a naive monkey learn the task by observing trained monkeys?

Our novel environment has the provision to allow multiple monkeys to freely move and access the touchscreen workstation. We therefore wondered whether a naive monkey could learn the same-different task by observing trained monkeys. This would further obviate the need for the TAT paradigm by allowing monkeys to learn from each other, and potentially reduce human involvement.

To explore this possibility, we performed social learning experiments on two naïve monkeys (M2 and M4). In each case, the naïve monkey was introduced along with a trained monkey (M1/M3) into the behaviour room, giving it the opportunity to learn by observation. Each day of social training for M2 involved three sessions in which he was first introduced into the behaviour room along with M1, then introduced together with M3, and finally a solo session. For M4 social training, we included a social session with M3 and a solo session. Neither monkey was acquainted with the setup at all prior to this. The results for each monkey are separately summarized below.

## Social learning of naive monkey M2

Here, naive monkey (M2) was intermediate in its social rank, with one of the trained monkeys (M1) being higher and the other (M3) being lower in rank. Initially, on each day of training session, M2 participated in two social training sessions: in the first session, it was introduced into the behavior room with M1. In the second session, it was introduced with M3. We also included a session in which M2 was allowed to attempt the task by himself with no other animal present. We used CCTV footage to retrospectively identify which monkey was doing the task on each trial during the social sessions. The data from the behavioral task together with information about monkey identity allowed us to quantify the performance each monkey separately during social training sessions. The results are summarized in *Figure 4*, and video clips of the key stages are shown in *Video 4*.

Video frames of key events are shown in *Figure 4A*. On Day 1, we observed interactions expected from the social hierarchy: M1 intimidated M2 and prevented any access to the workstation, and M2 did the same to M3. The M1-M2 dynamic remained like this throughout the social sessions. On Day 4, M2 pulled M3 into the behaviour room, and we observed a few trials in which M2 drank juice while M3 performed a few correct trials. By Day 5, M2 was observing M1 closely in the M1-M2 social sessions, and began to slide his hand to make a response in the M2-M3 social sessions. By Day 9, M2 was performing the task at chance level. By Day 13, there were no interactions between M1 and M2 (with M1 dominating throughout) and no interactions between M2 and M3 (with M2 dominating throughout). We therefore stopped the social sessions and began introducing M2 by himself into the behaviour room. From here on, M2 took eight more sessions to reach above-chance accuracy on the task. By the end of 29 sessions, M2 had achieved 91 % accuracy on the task. A more detailed description and analysis of social sessions is included in Appendix 2.

To quantify the social session performance of all monkeys, we plotted the overall accuracy of each monkey on trials in which they made a response to one of the choice buttons (*Figure 4B*). It can be seen that monkey M2 began to initiate trials correctly and make choice responses by Day 5, and his performance began to rise above chance by about Day 15. To further elucidate how M2 learned the

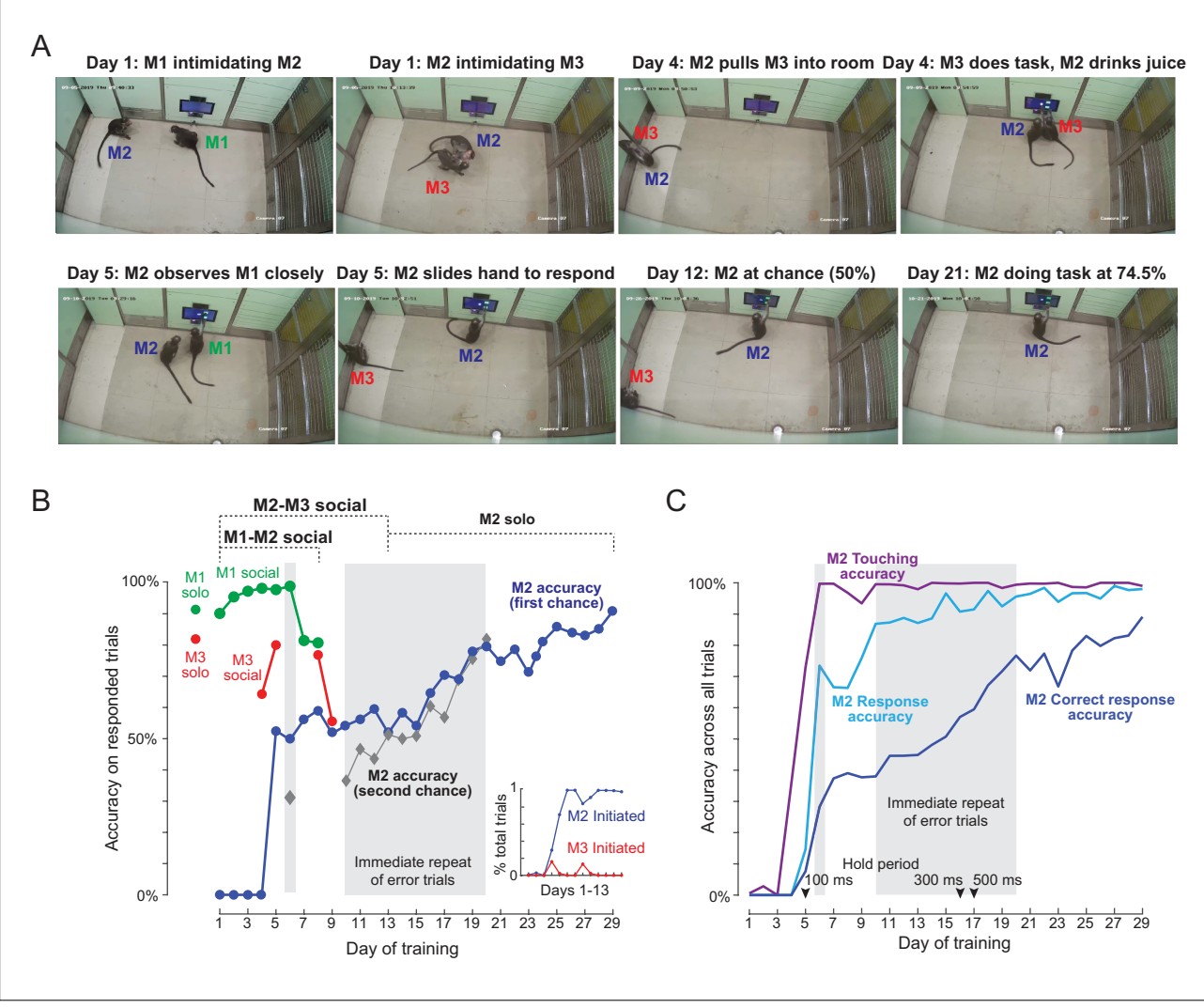

**Figure 4.** Social learning of naïve monkey M2. (**A**) Photos representing important stages of social learning for M2 by observing trained monkeys M1 and M3. Social rank was M1> M2> M3. See *Video 4*. (**B**) Accuracy in social training sessions (green-M1, blue-M2 and red-M3) across days. For each monkey, accuracy is calculated on trials on which it made a choice response. Shaded regions depict days on which error trials were repeated immediately, allowing monkeys to learn by switch their response upon making an error. M2 accuracy on such repeated trials is shown separately (*gray*). M1 and M3 accuracy prior to and during social sessions is shown by *red* and *green* dots (M1: 91%, M3: 82%). *Inset*: Percentage of all trials initiated by M2 (*blue*) and M3 (*red*) during M2-M3 sessions across 13 days of training. (**C**) Accuracy for monkey M2 for various types of response, calculated as percentage of all trials. *Touching accuracy (purple)*: percentage of all trials initiated by touching the hold button. *Response accuracy (cyan)*: percentage of trials where M2 touched any choice button out of all trials. *Correct response accuracy (blue):* Percentage of trials where M2 touched the correct choice button out of all trials. Shaded regions depict days on which error trials were repeated immediately without a delay. Arrow indicate days on which the hold time was changed.

The online version of this article includes the following figure supplement(s) for figure 4:

**Figure supplement 1.** Social learning for naïve monkeys M2 & M4.

same-different rule we separated his accuracy into trials with immediate repeat of an error ('second-chance accuracy') and trials without an immediately preceding error ('first-chance accuracy'). This revealed an interesting pattern, whereby M2 began to increase his second chance accuracy, presumably by switching his response upon making an error almost immediately after introducing immediate repeat of error on Day 10 (*Figure 4B*). Interestingly his first-chance accuracy only began to increase a few days later, from Day 16 onwards (*Figure 4B*). To evaluate how M2 learned various aspects of the task, we calculated several types of accuracy measures for each session: touching accuracy (percentage of trials initiated by touching the hold button), response accuracy (percentage of trials in which M2

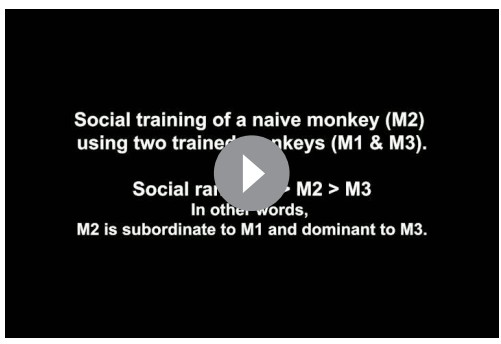

**Video 4.** Social learning of Monkey M2.
https://elifesciences.org/articles/63816/figures#video4

pressed either choice button) and finally correct response accuracy (percentage of trials where M2 touched the correct choice button). The resulting plot (*Figure 4C*), shows that M2 learned to touch by Day 2, respond to choice buttons by about Day 5, and began to make correct responses significantly above chance by Day 15.

## Social learning of naive monkey M4

The above results show that the naïve monkey M2 was able to learn the same-different task through social observation of trained monkeys as well as through solo sessions involving trial-and-error learning. To confirm the generality of this phenomenon, we trained a second naïve monkey M4 by letting him socially observe the trained monkey M3. Since we observed more interactions between M2 and M3 during social learning of M2, we selected the naïve monkey (M4) to be socially dominant over the trained monkey (M3). However, this social dominance reversed over time so that M3 became dominant over M4 by the start of the social sessions, and this trend also reversed at times across sessions.

On each day of social learning, we conducted three sessions: a solo session with only M3 performing the task, followed by a social session where M4 was introduced into the room with M3 already present, and finally a solo session with only M4. To summarize, M4 learned to touch correctly by Day 2, began to touch the choice buttons by Day five and his accuracy increased steadily thereafter reflecting continuous learning (*Figure 4—figure supplement 1*). However, a post-hoc analysis revealed that this improvement was primarily due to increase in second-chance accuracy with little or no change in first-chance accuracy. Thus, monkey M4 also demonstrated an initial phase of learning task structure, followed by a later stage of trial-and-error learning similar to the monkey M2. However the learning curve for M4 was unlike that seen for M2. Whereas M2 learned the same-different rule while also learning to switch his response on immediate-repeat trials, M4 only learned the suboptimal rule of switching his response on immediate-repeat trials. Nonetheless, M4 was successful at trial-and-error learning on this task, albeit with suboptimal learning. A descriptive analysis of the key events during social training of M4 is included in Appendix 2.

## How did monkeys learn during social learning?

The above observations demonstrate that both naïve monkeys (M2 and M4) learned the task in two distinct phases. In the first phase, they learned the basic structure of the task through social interactions and learning. By task structure we mean the specific sequence of actions that the animal has to perform to receive reward at chance levels: here, these actions involve holding one button until the test image appears and then touching one of the choice buttons afterwards and removing his hand from the touchscreen to initiate the next trial. By the end of this stage, both monkeys did not seem to be benefiting from socially observing or interacting with the trained monkey.

In the second phase, M2 learned the same-different rule all by himself through trial-and-error, by improving on both his first-chance and second-chance accuracy. M4 also showed learning on the task but unlike M2, his improvement was driven by his second-chance accuracy alone, indicating that he learned a suboptimal rule to improve his task performance. Nonetheless, in both monkeys, the social sessions naturally dissociated these two stages of learning.

## Discussion

Here, we designed a novel naturalistic environment with a touchscreen workstation with high-quality eye tracking that can be used to study visual cognition as well as natural and social behaviors in unrestrained monkeys. We demonstrate two major outcomes using this environment. First, we show that high-quality eye tracking can be achieved in unrestrained, freely moving monkeys working at the

touchscreen on a complex cognitive task. Second, we show that interesting novel behaviors can be observed in this environment: specifically, two naïve monkeys were able to learn aspects of a complex cognitive task through a combination of socially observing trained monkeys doing the task and solo trial-and-error. We discuss these advances in relation to the existing literature below.

## Relation to other primate training environments

Our novel naturalistic environment with a touchscreen is similar to other efforts (*Calapai et al., 2017*; *Tulip et al., 2017*; *Berger et al., 2018*), where the common goal is a seamless behavior station to enable training monkeys within their living environment. However, it is unique and novel in several respects.

First, we were able to achieve precise monitoring of gaze in unrestrained macaque monkeys. While viable gaze tracking has been reported in unrestrained large animals, there are technical challenges in achieving this with unrestrained macaque monkeys, whose small size results in an elevated line of sight for any eye tracker placed at arm's length. To our knowledge, this is the first report of accurate eye tracking in unrestrained macaque monkeys interacting at close quarters with a touchscreen. This is an important advance since such gaze signals are required for any complex cognitive tasks involving visual stimuli. We overcame this challenge through two innovations: (1) designing a juice spout with a chin rest that essentially enabled monkeys to achieve a highly stereotyped head position while performing the task, with hand-holding grill and optional head frames for additional stability; and (2) splitting the eye-tracker camera and the IR illuminator, to allow IR light to illuminate the eyes from below, resulting in high-fidelity tracking. Second, unlike other facilities where the touchscreen workstation is an add-on or housed in a separate enclosure (*Evans et al., 2008*; *Mandell and Sackett, 2008*; *Fagot and Paleressompoulle, 2009*; *Fagot and Bonté, 2010*; *Calapai et al., 2017*; *Claidière et al., 2017*; *Walker et al., 2019*), our touchscreen is mounted flush onto a modular wall that enabled social observation by other monkeys, which in turn enabled novel social interactions such as those described here. Third, we demonstrate that monkeys can be group-housed even with safe perches out of reach from humans, yet it is possible to isolate each animal individually and give it access to the touchscreen workstation (see Materials and methods).

## Social learning vs automated training

We have found that naïve monkeys can learn a complex cognitive task through a combination of observing other trained monkeys and by solo trial-and-error. An extreme interpretation of this finding is that only one animal needs to be trained through TAT and other animals can learn from it through social observation and solo trial-and-error. A more reasonable interpretation is that this approach could either work partially in many animals, or entirely in a few animals. Either way, it could result in substantial time savings for human experimenters by allowing more animals to be trained in parallel and minimize manual interventions or even reduce the time required in automated training.

Do monkeys take less time to learn socially as compared to an automated training regime? This question is difficult to answer conclusively for several reasons: (1) training progress is not directly comparable between social and automated training (e.g. automated training involves learning to touch, hold, making response etc. which are absent in the social training); (2) There could be individual differences in learning and cognition as well as relative social rank that confound this comparison (*Capitanio, 1999*); and (3) it is possible that monkeys could learn slower/faster in a different automated or social training protocol.

Keeping in mind the above limitations, we nonetheless compared the total times required for automated and social training times using two metrics: the number of sessions required to learn task structure and the number of sessions required to learn the same-different rule. For monkeys M1 & M3, which were on automated training, both learned task structure in 34 sessions and learned the same-different rule after 86 sessions. These training times are comparable to a recent study that reported taking 57–126 sessions to train animals on a simpler touch, hold and release task (*Berger et al., 2018*). By contrast, for monkeys M2 & M4, which underwent social training, both M2 & M4 learned task structure in 9 sessions and M2 learned the same-different rule after 25 sessions, whereas M4 learned a suboptimal rule instead. Thus, in our study at least, social learning was much faster than automated training.

In practice, we propose that one or two animals could be trained through automated approaches, and then the larger social group (containing the trained animals) could be given access to socially observe and learn from the trained animals. This approach could help with identifying the specific individuals that are capable of socially learning complex tasks - an interesting question in its own right.

### Insights into social learning

Our finding that naïve animals can learn at least certain aspects of a complex task through social observation is consistent with reports of observational learning in monkeys (*Brosnan and de Waal, 2004*; *Subiaul et al., 2004*; *Meunier et al., 2007*; *Falcone et al., 2012*; *Monfardini et al., 2012*), and of cooperative problem solving and sharing (*Beck, 1973*; *de Waal and Berger, 2000*). However, in these studies, naïve animals learned relatively simple problem-solving tasks and did not have unconstrained access to the expert animal to observe or intervene at will.

Our results offer interesting insights into how animals might efficiently learn complex cognitive tasks. In our study, learning occurred naturally in two distinct stages. In the first stage, the naïve monkeys learned the basic task structure (i.e., holding and touching at appropriate locations on the screen at the appropriate times in the trial) by socially observing trained monkeys, but did not necessarily learn the same-different rule. This stage took only a few days during social learning. This could be because the naïve monkey is socially motivated by observing the trained monkey perform the task and/or receive reward. In the second stage, the naïve monkeys showed little interest in social observation, often dominated the teacher due to their higher social rank, and began learning the task through trial-and-error. This stage took about two weeks for monkey M2, and we estimate it would take us a similar amount of time using an automated process such as TAT. Thus, the major advantage of social learning was that it enabled the naïve animal to learn the basic task structure from a conspecific, while learning the more complex cognitive rule by itself.

### Future directions: recording brain activity

Our naturalistic environment constitutes an important first step towards studying brain activity during natural and controlled behaviors. A key technical advance of our study is that we are able to achieve high-quality eye tracking in unrestrained monkeys, which will enable studying vision and its neural basis in a much more natural setting, as well as studying the neural basis of complex natural and social behaviors. Many design elements described in this study (e.g. electromagnetic shielding, snout restraint to permit wireless implant maintenance, neural data acquisition systems and related computers) are aimed at eventually recording brain activity in this setting. However, we caution that recording brain activity still requires several non-trivial and challenging steps, including surgical implantation of microelectrodes into the brain regions of interest, ensuring viable interfacing with neural tissue and ensuring noise-free wireless recordings.

## Materials and methods

All procedures were performed in accordance with experimental protocols approved by the Institutional Animal Ethics Committee of the Indian Institute of Science (CAF/Ethics/399/2014 & CAF/Ethics/750/2020) and by the Committee for the Purpose of Control and Supervision of Experiments on Animals, Government of India (25/61/2015-CPCSEA & V-11011(3)/15/2020-CPCSEA-DADF).

### Animals

Four bonnet macaque monkeys (*macaca radiata*, laboratory designations: Di, Ju, Co, Cha; all male, aged ~7 years – denoted as M1, M2, M3, M4 respectively) were used in the study. Animals were fluid deprived on training days and were supplemented afterwards such that their minimum fluid intake was 50 ml per day. Their weight and health were monitored regularly for any signs of deprivation. In a typical session, animals performed about 400–500 trials of the same-different task, consuming about 80–100 ml in a one hour period after which we typically stopped training.

To quantify these trends for each monkey, we analyzed 50 recent sessions in which three monkeys (M1, M2, M3) were trained on either a same-different task or a fixation task on each day (number of same-different sessions: 44/50 for M1; 28/50 for M2 and 47/50 for M3). All three animals performed a large number of trials per session (mean ± sd of trials/session: 540 ± 260 trials for M1, earning 104 ±

50 ml fluid; 574 ± 209 trials for M2, earning 94 ± 48 ml fluid; 395 ± 180 trials, earning 71 ± 30 ml fluid; mean ± sd of session duration: 41 ± 25 min for M1; 45 ± 17 min for M2; 26 ± 16 min for M3). In all cases, sessions were stopped either if the animal showed no consistent interest in performing the task, or if it had consumed a criterion level of fluid after which it would compromise consistent performance on the next day. We did not give unlimited access to the touchscreen workstation, and as a result, do not yet know the level of engagement possible in these scenarios.

## Overview of naturalistic environment

Our goal was to design and construct a novel environment with an enriched living environment with controlled access to a behavior room with a touchscreen workstation, and provision for training on complex cognitive tasks and eventual wireless recording of brain signals.

In primate facilities where monkeys have freedom of movement while interacting with behavior stations, the major differences typically lie in the placement of the behavior station relative to the living room, mode of interaction while monkeys perform tasks and the degree to which the animal's behavior could be observed by other monkeys. The simpler and more common approach has been to install the behavior station directly in the living room either on the walls (*Rumbaugh et al., 1989*; *Crofts et al., 1999*; *Truppa et al., 2010*; *Gazes et al., 2013*; *Tulip et al., 2017*; *Butler and Kennerley, 2019*) or in an adjacent enclosure where a single subject can be temporarily isolated (*Evans et al., 2008*; *Mandell and Sackett, 2008*; *Fagot and Paleressompoulle, 2009*; *Fagot and Bonté, 2010*; *Calapai et al., 2017*; *Claidière et al., 2017*; *Walker et al., 2019*). Although the former approach is easiest to implement and can let multiple monkeys interact with the behavior station, it can be challenging to prevent a monkey from getting distracted from other events in its living environment and to isolate individual monkeys for assessments. In contrast, the latter approach is better suited to control for disturbances in the living room but with the caveat that it has commonly been designed for use by one monkey at a time and thus precludes studying interesting behaviors where multiple monkeys can interact with the behavior station. An interesting recent approach is to use RFID technology to identify individuals that interact with the touchscreen (*Fagot and Paleressompoulle, 2009*; *Fagot and Bonté, 2010*).

Here, we combined the best of both approaches to create a single large naturalistic group housing area connected to a behavioral testing room through two intermediate rooms (*Figure 1A*). This allowed us to sequester the desired animal and send it into the behavior room for training or allow multiple animals to observe interesting social dynamics while they interact with tasks in the behavior room.

Our approach can be a practical blueprint for other monkey facilities who wish to implement an enriched living and behavior environment in their own larger or smaller spaces. To this end, we have included a detailed description and specifications of various architectural, electrical and mechanical components in our environment.

## Naturalistic group housing

We commissioned an environmental arena meeting our requirements which can house a small number of animals (3–6 monkeys). Monkey-accessible areas were separated from human-accessible areas using solid high-pressure laminate panels (HPL), toughened glass or stainless-steel mesh partitions (*Figure 1A*). The entire environment was designed by a team of architects and engineers (Opus Architects & Vitana Projects) using guidelines developed for NHP facilities (*Röder and Timmermans, 2002*; *Buchanan-Smith et al., 2004*; *Jennings et al., 2009*). We incorporated ample opportunities for the monkeys to interact with the environment and used natural materials wherever possible. We provided two perches at above 2 m elevation made of wooden beams on a stainless steel frame (*Figure 1B and C* top), repurposed tree trunks as benches, and a dead tree as a naturalistic feature for climbing and perching. Cotton ropes were hung from the taller elements for swinging and playing. We also included a stainless steel pendulum swing for playing.

To prevent tampering and to ensure safety, all electrical components like roof lights and closed-circuit television (CCTV) cameras were enclosed with stainless-steel and toughened glass enclosures (*Figure 1C*, bottom). None of the structural and mechanical elements had sharp or pointed corners or edges. This room as well as other monkey-accessible areas described below were provided with a constantly replenished fresh air supply and exhaust ventilation. To keep unpleasant odors under control

and to provide foraging opportunities for the monkeys, the floor of the living room was covered with a layer of absorbent bedding (dried paddy husk and/or wood shavings) that was replaced every few days.

Compared to the older living area for monkeys (stainless steel mesh cages), the naturalistic group housing area is much more spacious (24 times the volume of a typical 1m x 1m x 2 m cage) and includes a large window for natural light. The living room was designed for easy removal and addition of features (all features are fixed with bolts and nuts), thus allowing for continuous improvement in enrichment. The enriched living room was effective in engaging the animals as observed from heatmaps of their movements (*Figure 1D*). *Figure 1D* shows animal activity in a 7 min period, both with and without the presence of humans in the interaction area. Animals heavily interacted with the enriched environment, leading to an observable improvement in their behavioral and social well-being.

## Holding area and squeeze partition

From the group housing area, monkeys can approach the behavior room containing the touchscreen workstation (*Figure 1I*, touchscreen monitor for visual tasks and response collection) through a passageway (*Figure 1H*). The passageway is divided into two parts, a holding area and a squeeze room (*Figure 1E–F*). The holding area is adjacent to the group housing area and is designed to be employed when isolating an animal when required. A log bench was provided as enrichment in the holding area along with windows with natural light.

In the squeeze room, the back wall can be pulled towards the front to restrain the animals for routine tasks like intravenous injections, measurement of body temperature, closer physical inspection by the veterinarian, etc. The back wall is attached to grab bars in the human interaction room (to push and pull it) and a ratchet system (*Figure 1G*) to prevent the monkey from pushing back. This enables an experimenter to squeeze and hold the back wall in position without applying continuous force, allowing them to focus on interacting with the animal and minimize its discomfort.

All monkey-accessible rooms were separated by sliding doors that can be locked (*Figure 1G*, *bottom*) to restrict a monkey to any given room. Ideally, all the sliding doors could be left open, and monkeys can move freely across these rooms. In practice, to train individual animals, we often would shepherd the desired animal into the behaviour room by sequentially opening and closing the doors to each enclosure. We also incorporated trap doors to bring the monkeys out of each enclosure for the purposes of maintenance, relocation, or for other training purposes (*Figure 1E*). These trap doors allow for positioning a transfer cage or a traditional monkey chair into which the animal can be trained to enter.

## Animal training

The design of the naturalistic group housing room relinquishes a large degree of control by the experimenters. For instance, monkeys in this environment could easily opt out of training by perching at a height. They may never enter the holding area even on being induced by treats from the experimenters. A dominant monkey could potentially block access to subordinate monkeys and prevent them from accessing the behavior room. In practice, these fears on our part were unfounded. Initially during fluid deprivation and subsequently even without deprivation, monkeys would voluntarily approach the holding area when induced using treats by the experimenters and often even without any inducement (e.g. training sessions missed during a six month period: 6 % i.e. 6/101 sessions for M1; 0 % i.e. 0/101 sessions for M2; 4 % i.e. 3/79 sessions for M3). Once the animals are sequestered in the holding area, we would separate the desired animal by offering treats in the squeeze partition while simultaneously offering treats to the other animal in the holding area. This approach allowed easy separation of individuals even when one animal is trying to block access of the other. In the rare instances when the undesired animal moved into the squeeze partition, we would take it out into a conventional primate chair or transfer cage and put it back into the group housing area.

## Snout restraint

We also used standard positive reinforcement techniques to train animals to enter conventional primate chairs for maintenance of future wireless neural implants. To hold the head temporarily still, we devised a novel 3D-printed snout restraint (*Figure 2—figure supplement 3C*) that could be mounted on the flat portion of the primate chair, and slid forwards to temporarily immobilize the snout

(and therefore, the head). We trained monkeys to accept treats and juice through the snout restraint. We found that animals easily tolerate being restrained for upto 10–15 minutes at a time, and are able to drink juice and eat small treats without any sign of discomfort. This duration is long enough to any cleaning or maintenance of their brain implant. This novel snout restraint avoids the traditional solution of a surgically implanted head-post, at least for the limited durations required for our purposes. It is similar in spirit to the reward cones reported recently for non-invasive head restraint (*Kawaguchi et al., 2019*). We propose that our snout restraint could be a viable non-invasive alternative to head-posts in many other scenarios as well.

## Behavior room overview

The behavior room contains a touchscreen workstation on the wall separating it from the control room (*Figure 1A*). The workstation consists of a touchscreen monitor and juice spout (*Figure 1I*) mounted on high-pressure laminate (HPL) modular panels. These panels are mounted on stainless steel channels which allow for easy repositioning or swapping as required. The same panels also covered all other walls of the behavior room. All panels contained two identical HPL boards with a thin copper sheet sandwiched in between, and were electrically connected using jumper cables on the control room side. This paneling was done to shield the behavior room from electromagnetic interference that could potentially interfere with neural recordings. We confirmed the efficacy of the electromagnetic shielding by comparing signal quality in the control room with the behavior room (*Figure 2— figure supplement 2*). A detailed system diagram with technical details of all components required to record behavioral and neural data is given in *Figure 2—figure supplement 1*.

## Behavior room: touchscreen workstation

We affixed a commercial grade 15" capacitive touchscreen monitor from Elo Touch Solutions Inc (1593L RevB) to the modular panels at the behavior station (*Figure 2A and B*). The height of the monitor from the floor was chosen such that the center of the screen lined up with the eye-height of a monkey sitting on the floor in front of the behavior station. This display supported a resolution of 1,366 pixels by 768 pixels with a refresh rate of 60 Hz and the polling rate of the integrated projected-capacitive touch panel was ~100 Hz. The stimulus monitor and a second identical monitor (backup/observation unit located in the control room) were connected to a computer running the NIMH MonkeyLogic (*Hwang et al., 2019*) experiment control software (running on MATLAB 2019a). Digital input and output of signals was facilitated by a National Instruments PCI-6503 card and BNC-2110 connector box combination (DIOxBNC).

Above and below the monitor on the behavior station were two acrylic window openings (17.7 cm tall by 22.8 cm wide). We evaluated many transparent media including plate glass, high refractive index corning glass, reinforced glass as well as transparent polycarbonate. We evaluated these media using a simple setup with a model head. We found clear acrylic to be the best media for the transparent windows, by contrast to the other options which had either internal and surface reflections (plate/corning glass) or high attenuation of infra-red light (reinforced glass). Acrylic also offered better mechanical strength and scratch resistance compared to polycarbonate. These transparent acrylic windows enabled us to position a commercial infrared eye-tracker camera (ISCAN Inc, ETL 300HD, details below) above the monitor and an IR illuminator below the monitor (*Figure 2A and B*). We also placed two synchronized network camera (frame sync-pulse recorded in NIMH ML through DIOxBNC) above and below the monitor. We fine-tuned the relative placement of our binocular eye-tracker and synchronized network cameras to observe fine-grained eye movements as well as head and body pose of our animals as they perform different visual matching tasks (*Figure 2C*). A photodiode was also placed on the touchscreen (*Figure 2B*) to measure the exact image onset times.

## Behavior room: juice spout and head restraint

Because monkeys had to sip juice from the reward arm, this itself led to fairly stable head position during the task. To further stabilize the head, we designed modular head frames at the top of the reward arm onto which monkeys voluntarily rested their heads while performing tasks (*Figure 2— figure supplement 3*). We formed a variety of restraint shapes with stainless-steel based on 3D scans of our monkeys with progressively increasing levels of restriction (*Figure 2—figure supplement 3*). Positioning their heads within the head restraint was not a challenge for the monkeys and they

habituated to it within tens of trials. We also iterated on the structure of the reward arm, head restraint and fabricated custom attachments (hand grill, *Figure 2A*) that allow the monkey to comfortably grip at multiple locations with its feet and with the free hand and this in turn greatly reduced animal movement while providing naturalistic affordances on the reward arm (*Figure 1H*, right most panel).

The reward for performing the task correctly was provided to the monkey as juice drops delivered at the tip of a custom reward delivery arm (*Figure 2A–B*; *Figure 2—figure supplement 3*). This reward arm was a 1″ width hollow square section stainless steel tube. Concealed within it are two thin stainless-steel pipes – a juice pipe for delivering the juice to the monkey and a drainpipe to collect any remaining juice dripping from the juice pipe. The juice was delivered using a generic peristaltic pump on the pipe connecting the juice bottle to the end of the juice pipe in the control room. This pump was controlled by a custom voltage-dependent current driver circuit printed to a PCB (*Figure 2—figure supplement 2*) which in turn is controlled through a digital signal from NIMH MonkeyLogic via the DIOxBNC board. The reward arm was mounted on a linear guide which allowed us to adjust the distance of juice pipe tip (near monkeys' mouth) and the touchscreen. As a result, we can passively ensure the monkey sat at a distance that enables it to give touch response without having to stretch their arms and gave a good field of view of the monkeys' face and body for the cameras.

## Behavior room: gaze tracking

Eye movements were recorded using a customized small form factor ETL 300HD eye tracker from ISCAN Inc, USA with optical lenses that enabled eye tracking at close quarters. The eye-tracker primarily consisted of an infrared monochrome video capture system that we oriented to get a field of view that covered both eyes of the animal when its mouth was positioned at the juice spout and the animal was in position to do trials. Although we initially kept both the eye tracker illuminator and camera adjacent to each other below the touchscreen, we were faced with a smearing of the corneal reflection of the illuminator on the edges of the cornea when monkeys made up upward gaze movement. We resolved this issue by splitting the relative positions of the IR illuminator (placing it below the touchscreen) and the IR sensitive camera (placed above the touchscreen; see *Figure 2*) of the eye tracker system to provide robust eye tracking across the range of eye movements within our task.

The ISCAN system offers a parameterizable eye-gate, which is in effect a rectangular aperture in the monochrome camera's field of view and restricts the search space of the pupil and eye-glint search routines in the ISCAN software algorithm. The pupil and eye-glint search are based on the area (minimum number of pixels) and intensity-based thresholds that can be manipulated using interactive sliders in ISCAN's DQW software. We modeled the raw eye-gaze signal as the horizontal and vertical signed difference between centroids of the detected pupil and eye-glint regions of interest. The raw eye signal was communicated in real time to the computer running NIMH ML through the DIOxBNC analog cables. This raw eye-signal was read into the NIMH ML software and got rendered in real time onto another monitor that displayed a copy of the visual stimuli shown on the monkey touchscreen, while the monkeys performed touch-based visual tasks.

We evaluated other commercial trackers but found limitations such as the need for semi-transparent hot mirror on the monkey side or a sticker to be affixed on the monkey forehead (EyeLink). Neither of these were practical options at the time of evaluation. We also found that other trackers popular for non-human primate research (Tobii X-120, Tobii Pro Spectrum) did not work as reliably for our monkeys, presumably due to species differences. Such species specific limitations of commercial eye trackers have been reported before (*Hopper et al., 2021*).

## Calibration of gaze data

NIMH ML has a feature to display visual cues at selected locations on a uniform grid that the monkey can either touch or look at and obtain the liquid reward. We trained our monkeys to look at and then touch these visual cues. Since monkeys typically make an eye movement while initiating and performing the reach and touch, we exploited this to first center the raw eye signal with respect to the center of the screen and subsequently obtain a coarse scaling factor between changes in the raw eye signal and corresponding changes in the on-screen location. In this manner, we obtained a rough offset and scaling factor that maps the raw eye gaze signal with the on-screen locations of the monkey touch screen.

We then ran calibration trials where four rectangular fixation cues were presented in random order. The animal had to look at each fixation cue as and when it was shown, all the while maintaining hold on a button on the right extreme portion of the screen. The animal received a liquid reward at the end of a complete cycle of fixation cues for correctly maintaining fixation throughout the trials. These calibration trials provided us with pairs of raw eye-gaze (x, y) observations that corresponded to known locations on the touch screen. We then used linear regression to learn a transformation between the raw eye-data to touchscreen positions. We used these session-wise calibration models to transform eye-data if a higher degree of accuracy was required than what is provided by the initial coarse offset and scaling of the eye-signal that we manually perform in the beginning of each trial. In practice, even the coarse centering and scaling of raw eye-data was sufficient for gaze-contingent paradigms where the monkeys had to either passively view successive stimuli in a fixation paradigm, or when they had to maintain gaze on the sample and test stimuli during the same-different tasks. Although linear regression was sufficient for our purposes, we note that biquadratic transformations might further improve gaze quality (*Kimmel et al., 2012*; *Bozomitu et al., 2019*).

### Animal activity analysis (Figure 1D)

We performed a motion heatmap analysis on the CCTV videos recorded from the play area using publicly available code (https://github.com/andikarachman/Motion-Heatmap; *Rachman, 2019*; copy available at our OSF data/code repository ). This analysis was helpful to visualize movement patterns over time and is performed frame by frame. On each frame, the background image is subtracted and thresholded to remove small motion signals. The result of the threshold is added to the accumulation image, and a colour map is applied. The colour map is overlayed on the background image to obtain the final output. We note that previous efforts have used color markers for activity and movement analyses (*Ballesta et al., 2014*), and more recently it has become possible to use markerless movement and pose tracking (*Mathis et al., 2018*).

### Gaze quality analysis (Figure 3 & Supplements)

We quantified the consistency of the mean gaze fixation during periods of fixation contingent behavior by plotting the relative probability of the mean fixations (within a trial) across trials in each session for each monkey. Briefly, we calibrated the raw eye-data using the calibration models built with data from calibration trials and segregated the data during the period of fixation contingency (from initial fixation acquisition to after inter stimulus interval or end of trial, for same-different and fixation tasks respectively). We took the mean fixation location within a trial and plotted the relative probability of the mean fixations across all trials in the session using the *histogram2* function provided in MATLAB with the normalization property set to 'probability'. Violin plots were based on code from Holger Hoffmann's Violin Plot programs (retrieved on June 30, 2021 from MATLAB Central File Exchange https://www.mathworks.com/matlabcentral/fileexchange/45134-violin-plot).

### Acknowlegements

We thank Sujay Ghorpadkar (Opus Architects), Anagha Ghorpadkar (Vitana Projects), Rikki Razdan & Alan Kielar (ISCAN), Assad & Mahadeva Rao (Fabricators), Ragav (Atatri), Akhil (Sri Hari Engineering) and Ajit Biswas & Venu Allam (CPDM IISc Smart Factory) for their excellent professional services with developing all custom components. We thank Mr. V Ramesh (Officer in-charge) and Ravi & Ashok (workers) from the Primate Research Laboratory (PRL) for their outstanding animal maintenance and care.

## Additional information

### Funding

| Funder | Grant reference number | Author |
| --- | --- | --- |
| Wellcome Trust DBt India Alliance | IA/S/17/1/503081 | SP Arun |

| Funder | Grant reference number | Author |
|---|---|---|
| ICMR Senior Research Fellowship | 3/1/3/JRF-2015/HRD-SS/30/92575/136 | Thomas Cherian |
| UGC Senior Research Fellowship | 816 /(CSIR-UGC NET DEC, 2016) | Jhilik Das |
| DST Cognitive Science Research Initiative | SR/CSRI/PDF-06/2014 | Harish Katti |
| Ministry of Human Resource Development, Government of India | Senior Research Fellowship | Georgin Jacob |

The funders had no role in study design, data collection and interpretation, or the decision to submit the work for publication.

## Author contributions

Georgin Jacob, Conceptualization, Data curation, Formal analysis, Investigation, Methodology, Software, Validation, Visualization, Writing – original draft, Writing – review and editing, Specific contributions: Conceptualised the new lab, worked with Opus Architects to finalize the design and Vitana Projects for the implementation, wrote MATLAB-based codes for behavioral training, oversaw design and fabrication of juice delivery systems, worked on all aspects of monkey training, wrote the manuscript with SPA and incorporated feedback from all authors; Harish Katti, Conceptualization, Data curation, Formal analysis, Investigation, Methodology, Software, Validation, Visualization, Writing – original draft, Writing – review and editing, Specific contributions: Conceptualised the new lab, worked with Opus Architects to finalize the design, conceptualized the reward arm and head restraints and oversaw fabrication, oversaw and coordinated steel-works, and prototyping and testing of fabricated products, worked with ISCAN in customizing the head-free eye-tracker, wrote MATLAB-based codes for behavioral training, worked on all aspects of monkey training, wrote parts of the manuscript and provided feedback on manuscript drafts; Thomas Cherian, Conceptualization, Data curation, Formal analysis, Investigation, Methodology, Software, Validation, Visualization, Writing – original draft, Writing – review and editing, Specific contributions: Conceptualized the reward arm and head restraints and oversaw fabrication, oversaw and coordinated steel-works, and prototyping and testing of fabricated products, worked with ISCAN in customizing the head-free eye-tracker, performed system integration and testing, wrote MATLAB-based codes for behavioral training, oversaw design and fabrication of juice delivery systems, worked on all aspects of monkey training, wrote parts of the manuscript and provided feedback on manuscript drafts; Jhilik Das, Conceptualization, Data curation, Formal analysis, Investigation, Methodology, Software, Validation, Visualization, Writing – original draft, Writing – review and editing, Specific contributions: Conceptualized the reward arm and head restraints and oversaw fabrication, oversaw and coordinated steel-works, and prototyping and testing of head restraints, snout restraints, 3d printing, wrote MATLAB-based codes for behavioral training, performed shield testing, worked on all aspects of monkey training, wrote parts of the manuscript and provided feedback on manuscript drafts; KA Zhivago, Conceptualization, Formal analysis, Investigation, Methodology, Software, Validation, Specific contributions: Conceptualised the new lab, worked with Opus Architects to finalize the design, designed, identified and procured equipment required for behavioural and neural data monitoring and recording, performed system integration and testing, wrote MATLAB-based codes for behavioral training, performed shield testing, provided feedback on manuscript drafts; SP Arun, Specific contributions: conceptualised the new lab, worked with Opus Architects to finalize the design and Vitana Projects for the implementation, conceptualized the reward arm and head restraints and oversaw fabrication, worked with ISCAN in customizing the head-free eye-tracker, designed, identified and procured equipment required for behavioural and neural data monitoring and recording, wrote the manuscript with GJ and incorporated feedback from all authors, Conceptualization, Formal analysis, Methodology, Project administration, Resources, Software, Supervision, Validation, Visualization, Writing – original draft, Writing – review and editing

## Author ORCIDs
Georgin Jacob  http://orcid.org/0000-0001-8262-0155
Harish Katti  http://orcid.org/0000-0003-4223-0325
Thomas Cherian  http://orcid.org/0000-0002-4910-6880

Jhilik Das (iD) http://orcid.org/0000-0002-5569-376X
KA Zhivago (iD) http://orcid.org/0000-0003-0327-1036
SP Arun (iD) http://orcid.org/0000-0001-9602-5066

### Ethics

All procedures were in accordance to experimental protocols approved by the Institutional Animal Ethics Committee of the Indian Institute of Science (CAF/Ethics/399/2014 & CAF/Ethics/750/2020) and by the Committee for the Purpose of Control and Supervision of Experiments on Animals, Government of India (25/61/2015-CPCSEA & V-11011(3)/15/2020-CPCSEA-DADF).

### Decision letter and Author response

Decision letter https://doi.org/10.7554/eLife.63816.sa1
Author response https://doi.org/10.7554/eLife.63816.sa2

---

## Additional files

### Supplementary files

• Transparent reporting form

### Data availability

All data required to reproduce the results in the study are available at https://osf.io/5764q/.

The following dataset was generated:

| Author(s) | Year | Dataset title | Dataset URL | Database and Identifier |
| --- | --- | --- | --- | --- |
| Jacob G | 2021 | monkeylabseries4 | https://osf.io/5764q/ | Open Science Framework, 5764q |

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

## Appendix 1

### Tailored automated training

Here we describe the Tailored Automated Training (TAT) paradigm we used to train naïve monkeys to perform a same-different task.

### Methods

#### Animals

M1 and M3 participated in the Tailored Automated Training. The animals were each provided a 45 minute period of access (session) to the behavior station with no fixed order of access. Training was conducted only if animals voluntarily moved to the behavior room. Animals were moved one at a time through to behavior room, closing partition doors behind them. If the animal was not willing to go forward to the behavior room, training was not done on that day and the animal was supplemented with 50 ml of water later in the day. Weight of the animals were checked twice a week and if any sudden drop in weight was measured the animal was given time to recover (by removing water restriction and pausing training).

#### Stimuli

For TAT, stimuli were selected from the Hemera Objects Database and consisted of natural and man-made objects with a black background to match the screen background.

#### Training

The aim of the TAT was to teach monkeys the temporal same-different matching tasks (SD task), a schematic of which is shown in *Figure 3A*. We employed TAT as a proof of concept to show that it is possible to achieve unsupervised training for animals on a complex same-different (SD) matching task. We automated the training by dividing the SD task into sub-tasks (stages) with further levels within each stage to titrate task difficulty. Animals progressed to successive levels and stages based on their performance (when accuracy on the last 50 attempted trials within a session was greater than 80%). Like recent automated training paradigms (*Berger et al., 2018*), we provided an opportunity to go down a level, if the animal performed poorly but we ultimately moved to a more stringent level progression where the animals were not allowed to slide back to an earlier level/stage. We started from a lower level only when the training was resumed after a long break, due to unavoidable circumstances like equipment failure or issues related to animal health. Overall, we find that the rate of learning depends on animal's underlying learning capability and the design of the automated training regime. Hence to achieve fastest learning rates, we optimized the level-wise difficulty of the automated design.

In general, the progression of task difficulty across levels and stages was selected such the animal could always perform the task at above-chance performance. Although we set out to train animals using a completely automated pipeline, we also wanted to ensure that both our naive animals could complete the learning process in full without drop out as is common in many automated regimes (*Calapai et al., 2017*; *Tulip et al., 2017*; *Berger et al., 2018*). We implemented a pragmatic approach, to intervene and tailor the training parameters at particularly difficult stages for so as to avoid the monkey dropping out of the training process entirely.

The SD task was divided into ten conceptual stages. A single parameter was varied across levels within a stage. The smallest unit of the TAT is a trial, but composition of each trial is dependent on the current level. Each trial started with the presentation of trial initiation button and trials were separated by a variable inter-trial interval (ITI). The duration of ITI depends on the outcome of the current trial (500 ms for correct trials; 2000 ms for incorrect trials). Provision was made to change some parameters quickly without aborting the experiment. The ITI and reward per trial were adjusted within a session based on animal's performance. We increased ITI to give another level of feedback when animals were showing very high response bias by pressing only one button or when the animals were satisfied with 50 percent chance performance.

Liquid juice reward was delivered after every correct trial. We started each session with 0.2 ml of juice reward per trial. Juice reward was increased for consistent behavior but never decreased within a session. The motive behind increasing the reward was to keep the motivation high when learning a new task as any kind of error done by the animal aborts the trial. Monkeys got two

distinct audio feedback tones: a high-pitched tone for correct response and a low-pitched tone for incorrect responses (including uninitiated, aborted or no response trials).

## TAT stages

### Stage-1 (Touch)

A green button (square) was presented on the touch screen where monkey had to touch for reward. Any touch outside was considered as error. There were two levels in this stage (Button size: 200 × 600 pixels in level 1.1 and 200 × 200 pixels in level 1.2). Center of the buttons were same as the that of the hold button in *Figure 3A*.

### Stage 2 (Hold)

The hold button was presented, and monkeys had to touch and maintain the touch within the button area until it was removed. Any touch outside the hold button was considered an error. There were thirty levels in this stage, in which hold time varied from 100 ms to 3 s in equally spaced intervals. M3 cleared all the levels but M1 was trained only up to a hold time of 2.6 s.

### Stage 3 (1-Response Button)

A temporal same different task with only correct choice button was presented. Choice buttons were green colored squares and were presented above and below the hold button for same and different choices, respectively. Image presentation sequence was same as that shown in *Figure 3A*. We had a wait to hold time for initiating the trial as 8000 ms, pre-sample delay time of 500 ms, sample-on time of 400 ms and post-sample delay of 400 ms. We reduced the time to respond in this level from 5 s to 400 ms in several steps (in 1000 ms steps till 1 s, 100 ms steps till 500 ms and 50 ms steps till 400 ms). Four image pairs formed from two images were used to construct the same different task.

### Stage 4 (2-Response Buttons)

In this stage the wrong choice button (also of similar dimensions and color to the hold button) was also displayed with brightness that increased from 0 to the maximum intensity (same as the correct choice button). This is a full temporal same different task with an intensity difference between correct and wrong choice buttons. Wrong button was introduced in ten steps with brightness scaled relative to the maximum intensity (scaling factor for each level: 0.2, 0.4, 0.5, 0.8, 0.85, 0.90, 0.925, 0.95, 0.975, 1). A scaling factor of 1 meant that there was no intensity difference between the choice buttons, and the monkey would have to use the visual cues (sample & test images) to perform the task. Time to respond was 800 ms and all other task parameters are same as stage 3.

### Stage-5 (Ad-hoc Strategies)

We introduced two new strategies (Immediate Repeat and Overlay) to facilitate same-different training. With the immediate repeat strategy, for every wrong trial, we repeated the same trial again with a lower reward (0.1 ml) for correct response. This allowed the animal to switch its response upon making an error. In the overlay strategy, we presented images of sample and test side by side blended on the correct choice button (blended image = α*image + (1-α)*choice button), where α is a fraction between 0 and 1. We started the first level of this stage by giving three kinds of additional information (Button intensity difference, Immediate Repeat and Overlay) to identify the correct response. As the levels progressed, we removed the cues slowly. First, we removed button intensity difference in six levels (scaling factor of wrong button intensity in each level: 0.2, 0.3, 0.5, 0.7, 0.9, 1). Second, we removed the overlay cue in 15 levels. (Blending factor α: 0.5, 0.4, 0.3, 0.2, 0.15, 0.1, 0.09, 0.08, 0.07, 0.06, 0.05, 0.04, 0.03, 0.02, 0.01,0). We removed the immediate repeat of error when blend cue reached $\alpha$ = 0.06.

### Stage-6 (Test Stimulus Association)

Stages 6, 7, 8 and 9 were based on a spatial version of the same-different task. In Stages 6 and 7, a new condition was introduced with overlay on correct response, and this happened on 50 % of trials in trial bag. The remaining trials were already learned conditions which were shown with no overlay. A level with overlay on correct response was repeated with a level without overlay. This spatial task differed from the temporal tasks in the position of the test image (shifted right or between sample and hold button) and sample ON time (sample image is presented till the trial ends). Each level introduced two new images through two specific image pairs (Images A and B are

introduced through trials AA and AB). The trials only differed in the test image, so the monkey can do the task only by associating a test stimulus to the correct choice button. In all, we introduced 20 new images and 20 image pairs across levels. Since we were presenting newly introduced image pairs more often (ratio of new image pairs to learned image pairs is 1:1), the monkeys could reach 80 % accuracy without attempting all learned image pairs. Hence, to check the monkey's performance on all learned image pairs, we created the last level with all 20 image pairs presented equally likely without cue.

### Stage-7 (Sample Stimulus Association)
In this stage we introduced image pairs formed from two images which differed in sample image (Images A and B are introduced through image pairs AA and BA but not AA and AB). In total we introduced eight new image pairs formed from eight images. All other experimental conditions were same as Stage-6.

### Stage-8 (Sample and Test Association)
Here we presented 16 image pairs selected from Stage-6 and Stage-7 together.

### Stage-9 (Spatial same-different task)
All possible image pairs from 20 new images were introduced in this level and this was done along with learned pairs (ratio of new pairs is to learned pairs is 1:1 with new pairs shown with choice button overlay). In next level overlay was removed and in subsequent levels the proportion of new image pairs were increased (this was done in two levels: 75:25 and 100:0). We tested the generalization introducing two new set of images (number of images in these sets: 20 and 100) in next two levels.

### Stage-10 (Temporal same-different task)
The task was switched to temporal from spatial SD task. In the first level we retained the sample image and test image location, but we turned off the sample image before presenting the test image. There was no delay between sample and test. Next level, the sample and test were spatially overlapping and the delay between sample and test were zero. In the subsequent levels the delay between sample and test were increased in steps (50 ms, 100 ms, 200 ms).

## Results
The complete trajectory of training for both M1 & M3 is depicted in *Appendix 1—figure 1* and are summarized below.

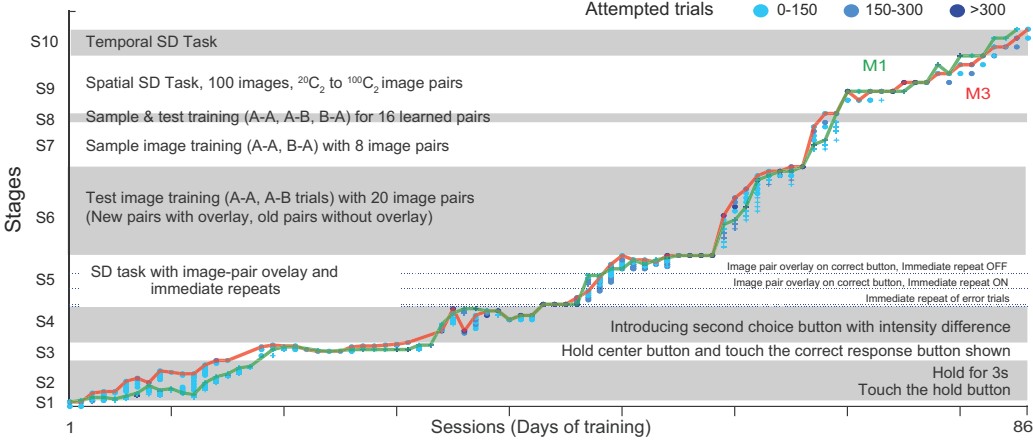

**Appendix 1—figure 1.** Tailored Automated training (TAT) on Same-Different task. The plot shows the progression of animals M1 and M3 through the ten stages of TAT. Each stage is further divided into levels with symbols corresponding to each monkey (plus for M1, circles for M3) and color

*Appendix 1—figure 1 continued*
indicating the number of trials attempted (0–150 trials: *light blue*, 150–300 trials: *cyan*, > 300 trials: *dark blue*). The lines indicate the maximum level reached by each animal in a given sessions (M1: *green*, M3: *red*).

Stage one was the touch stage: here monkeys had to touch a green square that appeared on the screen upon which it received a juice reward. Both monkeys cleared this stage in 1 day (*Appendix 1—figure 1*).

In Stage 2, monkeys had to hold their fingers on the green hold cue for increasing durations (100–3000 ms). The hold time was small initially (100 ms) so that monkeys would be rewarded for accidentally long touches and start to hold for longer periods. We trained monkeys to hold for longer periods (3 s) since this would be the hold time required eventually for the same-different task. Towards the end of this stage, we began to flash successive stimuli (up to 8 stimuli with 200 ms on and off) at the center of the screen while the monkey continued to maintain hold. Both monkeys took about two weeks to clear this stage (15 sessions for M1 to reach 2.6 s, 13 sessions for M3 to reach 3 s; *Appendix 1—figure 1*).

From Stage 3 onwards, monkeys started seeing a simplified version of the same-different task. Here we tried many failed variations before eventually succeeding. In Stage 3, they maintained hold for 500 ms, after which a sample image was shown for 400 ms, followed by a blank screen for 400 ms. After this a test image was shown at the center and the hold cue was removed, and a single choice button appeared either above (for SAME trial) or below (for DIFFERENT trial). To simplify learning, we used only two images resulting in four possible trials (either image as sample x either image as test). Monkeys had to release hold and touch the choice button within a specified time. Once monkeys learned this basic structure, we reasoned that reducing this choice time would force them to learn other cues to predict the choice button (i.e., the sample being same/different from test). However, this strategy did not work, and we discarded this strategy after 16 sessions (*Appendix 1—figure 1*).

In Stage 4, we introduced both choice buttons, but the wrong choice button had a lower intensity to facilitate the choice. Both monkeys quickly learned to select the brighter choice button. Here our strategy was to reduce the brightness difference to zero, thereby forcing the animals to learn the same-different contingency. Here too, monkeys kept learning to discriminate finer and finer brightness differences but failed to generalize to the zero brightness conditions. We discarded this strategy after 13 sessions (*Appendix 1—figure 1*).

In Stage 5, we tried several alternate strategies. These included immediate repeat of error trials (thereby allowing the monkeys to switch to the correct choice button), overlay of the image pair on the correct choice button (to facilitate the association of the image pair at the center with the choice buttons). While monkeys learned these associations correctly, they still did not generalize when these conditions were removed. On closer inspection, we observed that this was because they were looking only at the response button and not at the sample and test images. We discarded this strategy after 13 sessions (*Appendix 1—figure 1*).

In Stage 6, we further simplified the task by keeping the sample image identical in all trials, and varying only the test image (i.e., AA vs AB trials). We also simplified the task by showing the sample throughout, and then displaying the test image alongside the sample after a brief delay to facilitate comparison. We initially overlaid the image pair on the correct response button and eventually removed it based on performance. Monkeys cleared this level easily, and encouraged by this success, we introduced pairs of trials with new image pairs. In each level the old/learned pairs had no overlay (these were 50 % of the trials) and the new pairs had overlay (these were the remaining 50%). In this manner, we introduced 20 image pairs made from 20 unique images. Note that clearing this stage means that monkeys might have learned the full same-different concept or alternatively learned to associate specific test images to the "SAME" or "DIFFERENT" choice buttons. Monkeys cleared this stage in eight sessions (*Appendix 1—figure 1*).

In Stage 7, we attempted to nudge the monkeys towards a full same-different task. Here we used eight new images such that the test image was always the same in a given pair, but the sample image varied (i.e., AA vs BA trials). Monkeys cleared this stage in three sessions (*Appendix 1—figure 1*).

In Stage 8, we combined the trials from Stages 6 & seven in equal proportion (eight image pairs each). Monkeys cleared this stage in one session (*Appendix 1—figure 1*). However, it is still possible that they were doing this task by remembering sample or test associations with the corresponding choice buttons.

In Stage 9, we introduced all possible image pairs possible from 20 new images along with the previously learned image pairs and gradually reduced the proportion of the learned pairs. Both monkeys cleared stage easily (six sessions for M1, 5 sessions for M2), suggesting that they learned the concept of same-different. We further confirmed this by testing them on 100 new images, where sample and test images were chosen randomly from the $^{100}C_2 = 4,950$ possible sample-test pairs. Monkeys cleared this stage in 13 sessions (*Appendix 1—figure 1*).

In Stage 10, we transitioned to a temporal same-different task by reducing the temporal overlap between sample and test images, introducing a brief delay period, and then gradually moving the test image to the same position as the sample. Monkeys easily cleared this stage in four sessions (*Appendix 1—figure 1*).

# Appendix 2

## Social training

Social training of naïve monkey M2

### Animals

On each day of social training, M2 was involved in three sessions. First, he was introduced to the behaviour room with M1, then introduced with M3, and finally a solo session. M2 was group-housed with M1 and M3 from 9 months before start of social sessions, so their social hierarchy was observed to be M1> M2> M3.

### Stimuli

A set of 100 images of unique natural objects were used as stimuli. On Day 21 and Day 29, a new set of 50 images of unique natural objects were used to test the performance. All stimuli were presented after conversion to grayscale and the longer dimension of the images was always equated to 5.5° visual angle. Images were taken from the BOSS v 2.0 stimuli set (*Brodeur et al., 2010*; *Brodeur et al., 2014*) and from Hemera Photo Objects.

### Training

Temporal same-different task (stage 10 of TAT, *Appendix 1—figure 1*) was chosen for the social training sessions. Unlike TAT where an animal progressively attempts stages of the task until it is proficient in the full task, in social training sessions we investigated how a naïve monkey might learn the full task in the presence of trained peers (M1 and M3). Crucially, M2 can only get access to juice reward by responding when choice buttons are presented at the latter half of the trial.

Sessions were held on all mornings of the week except for Sundays and only if animals voluntarily moved to the behavior room (animals were herded two at a time through to behavior room, closing partition doors behind them). For instance, M3 did not come on Day three and Day 7; for these sessions, M2 was introduced alone into the behaviour room. If any animal did not come for a particular session, it was supplemented with 50 ml of water. Likewise, if the naïve or trained animal drank less than 50 ml juice during training, it was supplemented so that its total daily intake was 50 ml. Weight was monitored continuously as described earlier.

On each social session, we introduced M2 along with M1 (its superior in social rank) for 15–20 minutes or until M1 performed ~400 correct trials or 80 ml of juice. On the same day, we also introduced M2 with M3 (its subordinate in rank) for 45 minutes or until M2 received 60 ml of juice. Interestingly for few trials M2 and M3 cooperated (day 4: 35 trials, day 5: 14 trials, day 8: 96 trials and day 9: 10 trials; *Figure 4B* inset, *Appendix 2—figure 1*). M2-M3 session was for 45 minutes or until M2 received 50–60 ml of juice, whichever was earlier. Video recordings of both the sessions were done for subsequent coding of distinct behavioural episodes in these sessions.

Previous studies have established that animal learns more from peer's mistake (than from peer's success) and from own success (than own mistake) (*Monfardini et al., 2012*; *Monfardini et al., 2017*; *Isbaine et al., 2015*; *Ferrucci et al., 2019*). In a two-choice task, error reduces the preference of the choice made by the animal (*Monfardini et al., 2017*). In our case, the error signal is generated from multiple sources: breaking hold maintenance, incorrect response, and no response. We felt that maintenance of hold before the sample is shown is not crucial to task performance. Hence, we choose to make the task much easier and reduce errors by reducing the initial hold time down to 100 ms (on day 5) which reduced the hold maintenance time to 700 ms from 1.1 second. When the monkey started to get reward on 50 % of responded trials, we increase the initial hold time to be 300 ms on day 16 and 500 ms on day 17. After that the hold was 500 ms throughout the training. We modified inter-trial intervals (for correct and incorrect responses) and reward amount to keep M2 motivated to learn the task.

On Day 5, for few trials M2 was able to maintain the hold till the response buttons appeared. Then he dragged his hand below and touched the "different" response button (which was positioned at the bottom of hold button). He was able to obtain a reward on 50 % of the responded trials using this biased strategy. To discourage him from choosing only "different" button, on Day 6, we enabled immediate repeat of incorrect trials, so that an error trial was repeated immediately until he made a correct response. From Days 7–9, immediate repeat of error trial was disabled but on Day 10 we re-enabled immediate repeat of error trials to remove

response bias. Once M2's overall accuracy on responded trials (including immediate repeat of error trials) reached 80 % (Day 20) we disabled immediate repeat.

## Social session analyses

Since two monkeys were in the behaviour room during social sessions, we first identified which trial was done by which monkey by manually annotating the CCTV videos. Then for each monkey, we calculated accuracy on responded trials as a percentage of correct trials out of responded trials (*Figure 4B*). Accuracy could be of two types: First chance accuracy was calculated on all responded trials without including immediate repeat of error trials. Second-chance accuracy was calculated only on immediate repeat of error trials (after making an error, there were a stretch of same trial repeating, until the monkey made a correct response). For M1, repeat of error trials were not activated, and in case of M3, days when he did the task (day 4, 5, 8 and 9) immediate repeat of error trials were disabled. For M2-M3 session, we calculated percentage of trial initiated by M2 and percentage of trial initiated by M3, on total trials of that session (*Figure 4B* inset).

To understand the learning stages of M2 (*Figure 4C*), we calculated touching accuracy (percentage of total trial where M2 initiated the trial by touching), response accuracy (percentage of total trials in which M2 made a response) and correct response accuracy (percentage of total trials where M2 made a correct response). These three accuracies were calculated on total trials attempted by M2 alone (excluding the trials performed by M3).

## Social training of naïve monkey M4

### Animals

We introduced the naïve monkey M4 along with the trained monkey M3 for the social training. M4 and M3 were from the same social group, so M4 was pair-housed with M3 for 1 day before start of the social sessions. Their social hierarchy was observed to be M4> M3.

### Procedure

On each of social learning, we conducted three sessions: a solo session with only M3 performing the task, followed by a social session where M4 was introduced into the room with M3 already present, and finally a solo session with only M4.

### Stimuli and task parameters

All stimuli and task parameters were the same as the M2 social sessions except the following: (1) From Days 1–13, the stimulus set comprised 48 natural images divided into 24 blocks of 8 conditions (4 same and four different). On Day 14, this was changed to a single block of 2,550 trials created from 100 natural images, exactly as with the M2 social sessions; (2) From Days 1–13, the Hold period was 200 ms, and was reduced after that to 100 ms. (3) Error trials were set to delayed repeat on Days 1–8, ignore-on-error for Days 9–13, delayed repeat on Day 14, immediate-repeat from Days 15–33, and delayed-repeat on Days 34–39.

## Results

### Sequence of events during social learning of M2

How did M2 learn the task? Were there any key stages during this process? Since the social learning involved many uncontrolled one-time behaviours, we describe below both our descriptive observations together with quantitative analyses where possible of the entire social learning process.

On Day 1, we observed interactions expected from their social rank. In the M1-M2 session, M1 (being dominant) did the task and prevented M2 from approaching the touchscreen. In the M2-M3 session, M2 (being dominant) hogged the juice spout throughout and intimidated M3 whenever he approached the touchscreen (*Figure 4A*). This continued on Day 2, but M2 touched the hold button on a few trials though it did not progress through trial to get reward (*Figure 4C*, touching accuracy).

On Day 4, in the M1-M2 session, M2 watched M1 from a safe distance as before. But interestingly, in the M2-M3 session, M2 pulled M3 from the adjoining room into the behaviour room (see *Video 4*). Following this, M2 positioned himself in front of the juice spout, but also allowed M3 to access the screen. As a result, M3 performed a few trials while M2 received the juice

(*Figure 4A*). After this interaction, M2 initiated more trials by touching the hold button but still did not make further progress to get juice reward. These interactions are analysed quantitatively in *Appendix 2—figure 1*.

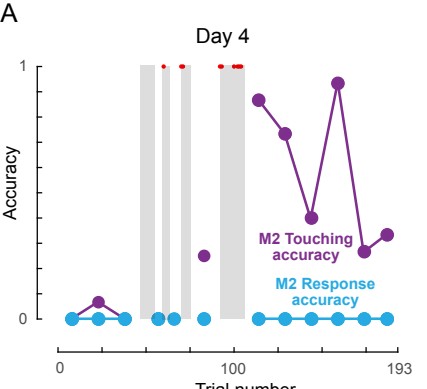

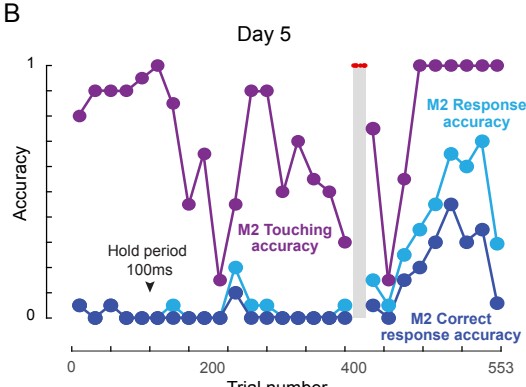

**Appendix 2—figure 1.** M2-M3 co-operation during social learning. Here we describe interesting social interactions between M2 & M3 during social training. To summarize, on Days 4 and 5, M2 was positioning himself in front of the touch screen, occupying the juice spout as usual, since M2 was dominant over M3. However, for some stretches, he allowed M3 to sit alongside closely such that M3 also had access of the touch screen. During these stretches, M3 performed the task for few trials (grey box), which included both correct and incorrect trials. Since M2 was occupying the juice spout, he got rewarded for these correct trials performed by M3. These interactions are detailed below. (**A**) Day 4, M2-M3 session: Shaded regions are showing trials where M2 and M3 co-operated in the task (M3 performed the task and M2 got juice). Red dots in shaded region are showing correct trials. The whole session is divided into non-overlapping bins (bin size is 15 trials except in the shaded regions). Each dot represents accuracy calculated on the total trials in that bin. *Touching accuracy:* percentage of trials initiated by M2. *Response accuracy:* percentage of responded trial (correct or incorrect) out of total trials. On this day, M2 was not touching the hold button much before the interaction trials (before trial 106), but after that M2 started initiating trials (*Figure 4—figure supplement 1A* ). He did not make any more progress. (**B**) Day 5, M2-M3 session: *Correct response accuracy*: percentage of total trials in which M2 made a correct response. Here bin size is 20 trials. All other conventions are same as (**A**). The arrow indicates the trial from which the hold time was changed (Day 1: 500 ms). From the beginning M2 was initiating the trials by touching the hold button but his response accuracy was very low (i.e. did not reach the two choices stage). He was able to maintain hold till response button appeared and made a response by dragging his hand through "Different button" for 13 trials before the interaction, out of which only four trials were correct. After this, M2 allowed M3 to perform the task for 14 trials (till trial 381) in which M2 received juice at a much higher rate (8 trials out of 14 were correct). After this interaction, M2's response accuracy increased (*Figure 4—figure supplement 1B*) and he started making correct response at chance level, although this was largely due to only making the (DIFFERENT response).

On Day 5, in the M1-M2 session, M2 watched M1 for long stretches. In the M2-M3 session, for a few trials, M2 maintained hold till the choice buttons appeared and ended up touching the lower button (corresponding to a DIFFERENT response) by dragging his hand down. M2 made four correct responses in this manner and received juice reward. After that, for a short stretch of trials, M2 allowed M3 to do the task (same as in Day 4) and M2 received the reward. M2 received the reward at a much higher rate (8 out of 14 trials of interaction, see *Appendix 2—figure 1*). After this M2 did not allow M3 to do any more trials, and his response accuracy and correct response accuracy increased, even though he continued to drag his hand through the DIFFERENT response button. On this day, the first chance accuracy of M2 was 53 % on responded trials (*Figure 4B*), though this was still a small proportion of all trials (7.6%, *Figure 4C*).

On Day 6, in the M1-M2 session, M2 watched M1 but only for a short duration. In the M2-M3 session, M2 started responding on more than 70 % of the trials and started making the SAME

response as well once we began immediate repeat of error trials (see Methods). Sometimes M3 was sitting beside M2, but M2 neither allowed M3 to do the task or showed any aggression to M3.

On Days 7 & 8, in the M1-M2 session, M2 watched M1 for a longer stretch, and M1 did not show any aggression even when M2 sat near M1. As in M1-M2 sessions, there was never any interaction between M1 and M2 (M1 dominated M2, and M2 watched M1 from a distance). We stopped M1-M2 sessions after Day eight as more interactions were happening in the M2-M3 session. On Day 7, M3 did not come for the task, thus in the M2-M3 session, M2 was attempting trials alone. On day 8, M3 was sitting closely with M2, and both M2 and M3 interacted for 96 trials in total (both did the task and sometimes shared reward, but mostly M2 occupied juice spout).

On Day 9, in the M2-M3 session, M2 allowed similar interaction for a very brief time, where M3 got to do the task (10 trials in total), and both were sharing reward. After this, M3 tried doing the task and occupying the juice spout by pushing M2 aside, but M2 showed his dominance. Overall, on this day, M3 sat beside M2 for a longer duration than Day 8. We did not see any improvement in M2's performance after the interactions on Day 8 & 9.

From Day 10 onwards, M2 did not allow M3 to attempt any more trials, while his task performance hovered around chance (*Figure 4B*). The duration for which M3 sat beside M2 also began decreasing after Day 9, and by Day 11, M3 was just roaming randomly in the room or sitting in the corner while M2 performed the task alone. After Day 13, we stopped the M2-M3 social sessions, and began introducing M2 by himself into the behaviour room (Day 14 onwards; *Figure 4B*). The M2-M3 interactions are summarized in *Figure 4B* (*inset*).

From Days 14–29, M2 was trained alone and learned the task by trial and error. We included an immediate repeat of error trials (Day 6 & Day 10–20), which allowed M2 to switch his response to the other choice button upon making an error. However, his accuracy on both the first-chance trials (i.e., trials without an error on the preceding trial) and on second-chance trials (i.e., on trials with an error on the preceding trial) increased monotonically, suggesting that he was continuously learning the concept of same-different and not just learning to switch on making an error (*Figure 4B*). By Day 25, M2 had attained an accuracy of 86%, meaning that he had learned the image same-different task.

## Sequence of events during social learning for M4
As before, we observed a number of interesting one-time events during social learning of M4, which we provide a qualitative description below.

On Day 1, we observed interactions expected from their social rank (M3> M4). M3 was doing the task, and M4 was observing the task from a safe distance. Both monkeys were not fighting inside the behavioural room. On Day 2, we observed similar behaviour by M3 and M4, but M4 started coming closer to M3 for watching the task. There was a long stretch (~5 minutes) of trials where both monkeys were accessing the screen together but M3 got all the reward.

During Days 3–5, M4 learned to initiate trials and began to get reward. M4 kept watching M3 for increasing periods, but M3 was unwilling to leave the juice spout opportunity to M4. During Days 6–8, M4 showed only a slight dominance over M3. On Day 6, this trend started reversing, and both M3 and M4 got more time with the screen alone. On Day 7, M4 showed complete dominance, occupying the screen more often and pushing M3 away from the juice spout.

On Day 8, the social session started with a fight between M3 and M4. After this fight, M3 again became dominant over M4, and M3 did all the trials with very high accuracy. There was no co-operation between M3 and M4 thereafter. On Day 9, M4 was not interested in doing the task in the social session. On Days 10–13, M4 showed interest in doing the task, sitting close to M3, but did not get a chance to do the task in the social session. During the solo session, M4 accuracy rose above chance. During this period M4 learned to avoid making touch and hold errors.

During Day 15–33 immediate repeat was on. While M4's overall accuracy began to improve steadily (*Figure 4—figure supplement 1F*), this improvement was largely due to his second-chance accuracy. In other words, he learned to switch his response after every wrong trial. Throughout this time, his first-chance accuracy remained at chance. Thus, M4 showed continuous learning but learned a suboptimal rule.

