## [Editor Report]

The manuscript describes a naturalistic experimental environment for training and testing macaque monkeys and for recording head-unrestrained eye movements. The utility of the setup is demonstrated through eye movement and social learning data during a cognitive (same-different) task. The authors conclude that this new environment provides a promising platform for studying cognitive and social behaviors, potentially in conjunction with wireless neurophysiological recordings in the future.

---

## [Decision Letter]

Thank you for submitting your article "A naturalistic environment to study natural social behaviors and cognitive tasks in freely moving monkeys" for consideration by *eLife*. Your article has been reviewed by three peer reviewers, and the evaluation has been overseen by a Miriam Spering as the Reviewing Editor and Chris Baker as the Senior Editor. The reviewers have opted to remain anonymous.

The reviewers have discussed the reviews with one another and the Reviewing Editor has drafted this decision to help you prepare a revised submission.

As you can see below, the editors have judged that your manuscript is potentially of interest. The reviewers emphasize that the paper reflects a hard-fought effort and commend you on describing a new research platform that has the capacity to transform how researchers approach the behavioral training of monkeys for some tasks. Whereas the reviewers are not asking for additional experiments to be conducted before the paper can be published, they nevertheless ask for extensive revisions. We would therefore like to draw your attention to changes in our revision policy that we have made in response to COVID-19 (https://elifesciences.org/articles/57162). First, because many researchers have temporarily lost access to the labs, we will give authors as much time as they need to submit revised manuscripts. We are also offering, if you choose, to post the manuscript to bioRxiv (if it is not already there) along with this decision letter and a formal designation that the manuscript is "in revision at *eLife*". Please let us know if you would like to pursue this option. (If your work is more suitable for medRxiv, you will need to post the preprint yourself, as the mechanisms for us to do so are still in development.)

Summary:

This manuscript describes a new experimental environment for training macaque monkeys to perform behavioral tasks. Using this facility, the authors trained freely moving macaques to perform a visual "same-different" task using operant conditioning, and under voluntary head restraint. The authors demonstrate that they could obtain reliable eye-tracking data and high-performance accuracy from macaques in this facility. They also noted that subordinate macaques can learn to perform basic aspects of the task by observing their dominant conspecifics perform the task in this facility. The authors conclude that this naturalistic environment can facilitate the study of brain activity during natural and controlled behavioral tasks.

The manuscript is broadly organized along three distinct lines of inquiry. First, the authors describe a customized living space for a small group of macaque monkeys. Second, the authors train two of these monkeys to perform a cognitive task in purpose-built room of the living enclosure. Third, the authors describe their experience training a third monkey to complete the cognitive task.

Essential revisions:

The main problem with the manuscript is that it is unclear in what way -- where along these three different topics -- the described environment represents a real methodological advance. It appears that the authors are currently not showing that the experimental environment is better than existing systems. Whereas the reviewers acknowledge that the manuscript describes a novel technology and therefore does not have to provide extensive research results, it would be important to clarify what the main advance is, and how the system can be validated. During their joint discussion, the reviewers and editors provided the following alternatives:

A) Social learning. If the advance is in this domain, then it needs to be substantiated. An anecdote is not enough. The authors would need to demonstrate that their system is really conducive to this form of learning, and this would require an entire study.

Specific comments with regard to social learning:

1) Throughout the manuscript, stating that the third monkey learned the task "merely by observing two other trained monkeys" is misleading. The naive monkey may have learned very important details about the cognitive testing set-up from observation. But the third monkey learned the task a unique behavioural shaping paradigm that included -but was not limited to- watching trained monkeys. The authors trained the third monkey on the cognitive task in the absence of the other monkeys, and do not show that the third monkey learned the specific cognitive task from watching other monkeys. Over-interpreting the anecdotal observations here hinders obfuscates what is novel and notable in this manuscript.

2) The authors repeatedly state that the third monkey learned the task faster than the previous two monkeys. It is quite difficult to parse exactly what the authors mean by this, and exactly what the data is that supports that claim.

3) The authors go on to state that M2 learned the "task structure" faster than M1/M3. However, "task structure" is not defined, so it is difficult for a reader to know precisely what was learned faster under social observation. Furthermore, the data showing that M2 learned the task structure faster than M1/M3 is not clear, and it is not known how M1/M3 learned the task structure in isolation. Description of which training steps may be aided by observation of trained monkeys must be clarified. The authors allowed M2 to observe M1 and M3 during initial familiarization of the experimental set-up, but it seems that observation may not have aided M2 in learning the complex same-different task at all.

4) Even though M2 may have learned the task structure faster than M1/M3, these observations are anecdotal and should not be over-interpreted. If there is a clear difference in the time to learn basic task structure, it may be due to social observation, but the authors should not favor that interpretation without considering alternatives as well. E.g., monkeys have widely varying personalities (see e.g. Capitanio 1999, Am J Primatology), and this has important implications for the curiosity, exploration behavior, and likelihood to accept and complete new challenges in training. To what extent could the differences in learning rate also be explained by these differences across these 3 monkeys? To what extend does the different training regimen in the task explain differences in learning rate across monkeys (e.g. M2 got two days of repeating correction trials, which significantly alters learning rates)?

5) The authors claim that it is easier to place a testing system into a separate cage then in the home cage. It remains unclear what this claim is based on. Motivation of animals in these social settings should be more difficult than in the home-cage environment. So, this is a potentially interesting result. It is also a conceptually important claim for the paper's logic, if the social setting should really be beneficial for training. But the claim needs to be substantiated.

B) If the advance is that of a low-cost system from which other labs should be able to profit, then a lot more information on cost and technical information for reproducibility should be provided, a behavioral guide for how to advance eye tracking etc.

Specific comments with regard to technological advance / eye tracking / neural recordings:

1) There is a vast literature in ethological settings where the gaze of nonhuman primates has been tracked using noninvasive methods that the authors do not acknowledge. Instead, authors state that most infrared eye trackers require head restraint (line 32), though this is demonstrably not the case. For review, see Hopper et al. 2020, Behav Res Methods.

2) The paper presents the testing environment consisting of different rooms. Compared to earlier work (e.g. Berger et al., 2018), the main innovation is the inclusion of an eye tracking system. Data supports the notion that this works in principle. But there is no analysis of data quality and accuracy. We also do not know whether the system works on every trial, or how often the eye is not detected or the tracker loses the signal.

3) The authors claim that natural behavior can be analyzed because a CCTV camera is mounted in the cage. There are no results or analyses to demonstrate that.

4) The authors mention neural recordings on multiple occasions, but do not show any. EM shielding is neither necessary nor new. Whereas the reviewers are not specifically asking for additional data, the authors need to rewrite sections referring to neural recordings if they do not provide any.

[Editors' note: further revisions were suggested prior to acceptance, as described below.]

Thank you for resubmitting your work entitled "A naturalistic environment to study social behaviors and cognitive tasks in freely moving monkeys" for further consideration by *eLife*. Your revised article has been evaluated by Chris Baker (Senior Editor) and a Miriam Spering (Reviewing Editor).

The manuscript has been improved but there are still some substantial issues that need to be addressed and would improve the manuscript further, as outlined below:

Before publication, all reviewers would like to see that some over-statements in interpreting the results are dealt with. In particular, some of these statements are related to whether monkeys M2 and M4 learned the complex cognitive task by social observation. The reviewers suggested during the post-review discussion that the part on social observation be moved into Supplementary Materials. The novelty of the presented method is also slightly overstated in places (see detailed reviewer comments below). Please be sure to address reviewer comments point-by-point.

*Reviewer #1:*

This revised manuscript is considerably improved in terms of its focus, rigour, and clarity. The authors undertook an immense amount of work through the revision process already, including training another naïve monkey and much more depth and detail included regarding the environment, training, and training outcomes.

I have included several recommendations for the authors. I feel strongly that these recommendations and suggestions should be addressed, but I don't think any of these points should preclude publication, and I'd trust the authors discretion in dealing with these suggestions.

1) In a few locations the authors still state that naïve monkeys learned the complex cognitive task by socially observing other trained monkeys. In my view, this is an over-interpretation that detracts from the manuscript, which I otherwise find to very interesting.

All of the following points are very clear in the revised manuscript: Monkey M2 learned the same-different task faster than monkeys M1 and M3. Monkeys M2 and M4 learned the basic task structure and how to interact with the touchscreen and lick spout while other monkeys were present. Monkey M2 learned the complex cognitive task while alone in the behavioural testing room. Monkey M4 did not learn the complex same-different task, but an alternate strategy to get enough juice to satisfy them for the session. Neither monkey M2 nor M4 learned the same-different task by observing M1 or M3; learning beyond the basic task structure and interaction with the screen happened when the monkeys were alone. These are all very clearly described in the results.

Therefore, statements like line 387 ("The above results show that a naïve monkey can learn a complex cognitive task by observing trained monkeys doing the task") are unfounded. The merits of the paper do not rely on monkeys learning a "complex cognitive task" through social learning, so I see no reason to include an over-interpretation results that are clearly explained in the text. The paragraphs starting on Line 445 and Line 509 are clear and measured. These are more accurate than the sentences on lines 397, 465, and the entire paragraph starting on line 501. I suggest that those parts of the text be amended.

2) In two locations in the revised manuscript, it is mentioned that the monkeys sometimes were not trained because they did not enter the behavioural testing area voluntarily (line 335, line 1329). It would be of interest for those that might want to replicate this type of facility to know the percentage of training days were missed for each monkey because they did not voluntarily enter the training room.

3) Line 70: This sentence suggests that the methods explain how experimenters isolated individual monkeys, but I didn't see any information about that beyond starting that a positive reinforcement training protocol was used. Given that some days monkeys did not come into the behavioural training room, I infer that there is not an easy way of isolating a specific animal from the group housing, particularly those animals of lower social rank. Whether there was a way to reliably get one specific monkey into the behavioural testing facility is not clear (particularly those of lower rank, who would defer to the higher ranked monkeys when treats are offered).

4) Readers might look at e.g. Figure 3, and interpret this to be the upper bound of eye tracking accuracy and gaze reconstruction possible in the testing environment described in the current manuscript. It might be of benefit to the authors and to the readers if the text explicitly stated that further optimizations could be performed but were not because they were not necessary for the present experiments. Given that this is a technical paper, these technical considerations could be discussed for the benefit of others who might be interested in adopting these techniques.

One important example: on line 811, the authors state that calibration was done using a linear mapping. Though this does not at all detract from the merits of this paper, it should be noted that this is not ideal for mapping raw eye data values to eye location. Biquadratic equations provide better estimates of gaze positions. Many papers have been published to this effect, but see https://www.ncbi.nlm.nih.gov/pmc/articles/PMC6721362/ for one example. Also, MonkeyLogic and the Eyelink system rely on the same biquadratic equation for their calibration routines; I'm uncertain about ISCAN.

5) Line 390: The authors state that M4 is dominant over M3, but the dynamic described below is clearly more complicated than that. This sentence initially confused me, and might confuse others.

6) Line 1196: There is no Figure S6. Please reference the correct figure number.

7) Line 1255: "M2" should actually be M3. M2 did not undergo TAT.

*Reviewer #2:*

The manuscript by Jacob et al. describes "A naturalistic environment to study social behaviors and cognitive tasks in freely moving monkeys". There is not much new to the individual components of this environment (more on this below), but this is the first time these components have been put together in this specific way. The main question in this review stage then is whether the manuscript presents a sufficient advance to warrant publication in *eLife*. The answer would be a clear yes for this reviewer if the new system would allow for fully automatic training inside the monkeys' living environment. This would constitute a major synergy between known components to a point where this combination would be a new system and something that many non-human primate labs would want to adapt. Alas, while there are elements towards such a system presented, this main advance has not been made yet. Therefore, the manuscript, before and after revision, appears fragmented and somewhat unfinished. Still, the description of a step in this direction might be of interest for readers of *eLife*. Furthermore, *eLife* establishes criteria for technology papers quite clearly:

"… authors will report substantial improvements and extensions of existing technologies. In those cases, the new method must be thoroughly compared and benchmarked against existing methods used in the field. Minor improvements on existing methodologies are unlikely to fare well in review." The current work falls into this category, and thus the onus is on the authors to really demonstrate this advance relative to the past literature. It is here where the manuscript, even after revision, falls short. While touting their own work as a "paradigm shift" or "exciting development", relevant past literature is glossed over and often not even discussed.

A case in point is the treatment of past work on eye tracking in macaque monkeys. The authors state that "there are relatively few studies showing this on macaque monkeys" and cite one reference, Hopper (2021). It might thus appear to a reader not particularly familiar with the subject that this is one example of these few studies, when really Hopper is a review article on "The application of noninvasive, restraint-free eye-tracking methods for use with nonhuman primates". Even a quick look into the article shows a great many studies cited and discussed in there. There is no discussion of any of the original work in this review, and there is no mentioning of the subject in the Discussion. This treatment of past literature is easily misleading and scientifically problematic. The authors need to take the past literature seriously and accurately represent the advance that they are making.

On the subject of eye tracking, the revision is moving in the right direction presenting and quantifying some eye tracking data. The authors describe their method as unrestrained. That is debatable, given that a chinrest is used, which would not qualify for unrestrained in human eye tracking for example. The interesting point here is that their subjects are moving to that chin rest by themselves, and one would wish that this advance would be highlighted. Quantification appears to be for a single session of a single subject only. This is not convincing. Even in a badly working system, one could find some good data. A quantification over more than one subject and multiple consecutive sessions would be necessary.

My third criticism is that claims of observational learning are overstatements of anecdotal observations. It would be better for the scientific quality of the paper, if it mentioned these observations as such, e.g. in supplemental information, focus on the enabling character of the technology for social learning, but not place much emphasis on these few observations (and avoid anthropomorphic interpretations).

*Reviewer #3:*

The manuscript describes a new "naturalistic" experimental environment for training and testing macaque monkeys on a popular cognitive task (delayed match-to-sample, also known as a "same/different" task). The manuscript demonstrates that: (1) the animals' eye movements can be monitored with sufficient precision in this environment; (2) the animals can be trained to perform this task with minimal human involvement; (3) the animals can learn faster by watching each other perform the task compared to being trained individually. The authors conclude that this new environment provides a promising platform for studying cognitive and social behaviors, potentially in conjunction with wireless neurophysiological recordings in the future.

The manuscript represents an important technical advance in that it demonstrates the feasibility of obtaining robust eye tracking data from monkeys housed in a naturalistic environment. The revised manuscript is an improvement on the initial submission, and likely of interest to researchers who wish to study monkey behavior in a richer and more dynamic setting compared to the traditional lab environment. Given this potential impact, the manuscript is a good addition to the field. Nonetheless, the manuscript can be further improved, as outlined below. Importantly, the authors might wish to re-consider their emphasis on the utility of this new environment for studying (1) social behavior and (2) monkey neurophysiology given that they have not pursued either of these directions fully.

Suggestions for improvement

1) The manuscript would benefit from the removal of the quote at the beginning.

2) The manuscript would be clearer if the authors refrain from using the word "hybrid". The authors use this word to mean "naturalistic" (as per the manuscript title), and the work "naturalistic" can be used throughout for improved clarity.

3) It seems inaccurate for the authors to emphasize the study of "social behaviors" in the manuscript title, especially since the data shown area all related to the cognitive "same/different" task. It would be more appropriate to highlight the future utility of using their naturalistic environment for studying social behaviors in the Discussion section.

4) Similarly, it seems inaccurate for the authors to emphasize the direct utility of their new environment for conducting wireless neurophysiological recordings because they have not yet demonstrated the feasibility of this approach. It would be more appropriate to highlight this point as a future direction in the Discussion section. In particular, these lines should be revised because retaining them would be misleading to the reader:

– Line 42: Here we designed a hybrid naturalistic environment with a touchscreen workstation that can be used to record brain activity…

– Line 50: We designed a novel naturalistic environment for recording brain activity…

– Line 460: Here, we designed a novel hybrid naturalistic environment with a touchscreen workstation that can be used to record brain activity

– Line 560: In sum, our environment represents an important first step in turning the traditional monkey neurophysiology paradigm on its head….

(5) The Discussion section would benefit tremendously from an additional paragraph about the many remaining challenges/limitations to neurophysiological recordings in this naturalistic environment. Otherwise, the reader might walk away thinking that achieving this next step of neural recordings is trivial when it is not.

---

## [Author Response]

Essential revisions:The main problem with the manuscript is that it is unclear in what way -- where along these three different topics -- the described environment represents a real methodological advance. It appears that the authors are currently not showing that the experimental environment is better than existing systems. Whereas the reviewers acknowledge that the manuscript describes a novel technology and therefore does not have to provide extensive research results, it would be important to clarify what the main advance is, and how the system can be validated. During their joint discussion, the reviewers and editors provided the following alternatives:

Thank you very much for this important point. In formulating this work, we too had extensive discussions about whether this manuscript is about social learning or about novel methodology. We strongly believe our study represent two key methodological advances: (1) highly accurate gaze tracking from unrestrained head-free animals, making it possible to study brain activity in both natural and controlled settings; and (2) monkeys can learn to perform complex cognitive tasks through social observation of trained monkeys. This too is an exciting methodological advance because many animals can now be trained through social observation of one trained animal, thereby saving months of tedious experimenter time that is being invested by primate labs worldwide for animal training. Both advances represent a win-win for science as well as animal welfare.

A) Social learning. If the advance is in this domain, then it needs to be substantiated. An anecdote is not enough. The authors would need to demonstrate that their system is really conducive to this form of learning, and this would require an entire study.

Thank you for this comment. We now report the results of social training of a second animal (M4). This animal also rapidly learned the task structure in a few days through social observation of another trained animal, and continuously improved its performance through trial-and-error. While the first monkey learned to switch its response on making an error and eventually learned the same-different rule, the second monkey only learned to switch its response and did not learn the same-different rule. Nonetheless both naïve monkeys showed clear learning of task structure and learning through trial-and-error, and their results show that learning of a complex task through social observation can happen in general. These results are now described in the main text and supplement.

Specific comments with regard to social learning:1) Throughout the manuscript, stating that the third monkey learned the task "merely by observing two other trained monkeys" is misleading. The naive monkey may have learned very important details about the cognitive testing set-up from observation. But the third monkey learned the task a unique behavioural shaping paradigm that included -but was not limited to- watching trained monkeys. The authors trained the third monkey on the cognitive task in the absence of the other monkeys, and do not show that the third monkey learned the specific cognitive task from watching other monkeys. Over-interpreting the anecdotal observations here hinders obfuscates what is novel and notable in this manuscript.

We agree that the naïve monkey did not learn the task entirely through social observation. That was never our claim. In fact, we have thoroughly analysed our social training sessions to parse out what the naïve monkeys learned socially and what they learned by themselves. We found that both monkeys initially learned the structure of the task through social observation after which they lose interest in the social interactions and learn the rule through trial-and-error. We have now clarified this throughout the text.

2) The authors repeatedly state that the third monkey learned the task faster than the previous two monkeys. It is quite difficult to parse exactly what the authors mean by this, and exactly what the data is that supports that claim.

Thank you for bringing up this point, this was indeed not clear in the manuscript. We have now detailed this clearly in the Results (p. 22).

3) The authors go on to state that M2 learned the "task structure" faster than M1/M3. However, "task structure" is not defined, so it is difficult for a reader to know precisely what was learned faster under social observation. Furthermore, the data showing that M2 learned the task structure faster than M1/M3 is not clear, and it is not known how M1/M3 learned the task structure in isolation. Description of which training steps may be aided by observation of trained monkeys must be clarified. The authors allowed M2 to observe M1 and M3 during initial familiarization of the experimental set-up, but it seems that observation may not have aided M2 in learning the complex same-different task at all.

From these concerns it look like several points need to be clarified:

1. Each day of social training for M2 involved two sessions in which he was first introduced into the behaviour room along with M1, then introduced together with M3, and finally a solo session. For M4 social training, we included a social session with M3 and a solo session. Neither monkey was acquainted with the setup at all prior to this.

2. By task structure, we meant the sequence of responses that the monkey has to make throughout the trial regardless of the same-different rule. In other words, even before learning the same-different rule, the monkey would have to learn to hold his hand on the screen to initiate the trial, keep holding throughout and touch one of the choice buttons that appear after the test stimulus is turned on. During this phase, we observed the naïve monkey

3. We can affirm that simply observing the experimental setup does not offer any advantage to a completely naïve monkey: when we first introduced M1 and M3 to our touchscreen setup before starting their automated training, they hardly interacted with the touchscreen or juice spout. Even if they did, they would not know what to do to even get reward.

The trained monkeys M1 and M3 were trained using the automated training approach (TAT) described in the section immediately preceding social training (Results, p 14).

4) Even though M2 may have learned the task structure faster than M1/M3, these observations are anecdotal and should not be over-interpreted. If there is a clear difference in the time to learn basic task structure, it may be due to social observation, but the authors should not favor that interpretation without considering alternatives as well. E.g., monkeys have widely varying personalities (see e.g. Capitanio 1999, Am J Primatology), and this has important implications for the curiosity, exploration behavior, and likelihood to accept and complete new challenges in training. To what extent could the differences in learning rate also be explained by these differences across these 3 monkeys? To what extend does the different training regimen in the task explain differences in learning rate across monkeys (e.g. M2 got two days of repeating correction trials, which significantly alters learning rates)?

We now acknowledge that automated and social training cannot be directly compared, and we now acknowledge this in the Results (p. 22). Even if learning through social observation takes as long as automated training, it would still result in significantly less experimenter involvement during training. We now acknowledge these points in the Discussion (p. 25).

5) The authors claim that it is easier to place a testing system into a separate cage then in the home cage. It remains unclear what this claim is based on. Motivation of animals in these social settings should be more difficult than in the home-cage environment. So, this is a potentially interesting result. It is also a conceptually important claim for the paper's logic, if the social setting should really be beneficial for training. But the claim needs to be substantiated.

A major concern in group housing is that individual animals cannot be easily isolated for testing or training. We have overcome this problem by creating spaces for guided movement of individuals or subgroups, and show that it is possible to isolate and train individual animals within this environment. This avoids the need for any artificial restraint systems like monkey chairs, poles etc. We now clarified this in the Methods (p. 29).

B) If the advance is that of a low-cost system from which other labs should be able to profit, then a lot more information on cost and technical information for reproducibility should be provided, a behavioral guide for how to advance eye tracking etc.

This was our goal also. To this end, we have included all possible technical information, such as commercial product model numbers, design diagrams with dimensions, detailed information about how to achieve good eye tracking, etc. Many of these items require custom design based on the general principles described here. We are happy to include any further information that the Editors or Reviewers think will be useful to the broader neuroscience community. We look forward to your suggestions.

Specific comments with regard to technological advance / eye tracking / neural recordings:1) There is a vast literature in ethological settings where the gaze of nonhuman primates has been tracked using noninvasive methods that the authors do not acknowledge. Instead, authors state that most infrared eye trackers require head restraint (line 32), though this is demonstrably not the case. For review, see Hopper et al. 2020, Behav Res Methods.

Thank you for pointing us to this reference. While it is true that gaze tracking has been reported in unrestrained animals, the vast majority of these studies are on large animals whose body dimensions are similar to humans, which enable commercial eye trackers to work. There are relatively few studies on unrestrained macaque monkeys. Moreover, the small size of these monkeys implies an elevated line of sight for any eye tracker placed at arm’s length of the animal, making tracking much more difficult. We in fact evaluated a number of commercially available eye trackers before going into a custom design cycle with our current eye tracking system (from ISCAN, Inc). We have now acknowledged these points and expanded upon them in the Introduction (p. 4), Discussion (p. 24) and Methods (p. 35).

2) The paper presents the testing environment consisting of different rooms. Compared to earlier work (e.g. Berger et al., 2018), the main innovation is the inclusion of an eye tracking system. Data supports the notion that this works in principle. But there is no analysis of data quality and accuracy. We also do not know whether the system works on every trial, or how often the eye is not detected or the tracker loses the signal.

Thank you for raising this point. We now include gaze traces from both monkeys (M1 and M3) during both a same-different task as well as a fixation task. Our supplementary videos show how we have achieved stable head position and gaze tracking through the juice spout design. We have four synchronized video cameras synchronized to the behavioural trials, and on careful review, we find that the eye tracking is lost only when the monkeys looked away from the screen. We now included these in the Results (p. 10) and Figure 3 supplements.

3) The authors claim that natural behavior can be analyzed because a CCTV camera is mounted in the cage. There are no results or analyses to demonstrate that.

On the contrary, our claim is that natural behaviour can be analysed using the CCTV cameras placed throughout the environment, and four video cameras placed near the touchscreen. Eventually we hope to record brain activity during these natural behaviours, enabling exciting insights. In this study, our clearest example of natural behaviour is the social learning experiments in which a naïve monkey was able to learn by socially observing a trained monkey perform the task. We used CCTV cameras to correctly identify the monkey performing the task during social sessions with both animals present in the room (Results, p. 15-16).

4) The authors mention neural recordings on multiple occasions, but do not show any. EM shielding is neither necessary nor new. Whereas the reviewers are not specifically asking for additional data, the authors need to rewrite sections referring to neural recordings if they do not provide any.

Our facility is built for wireless neural recordings, and is fully functional. Unfortunately, our plans for wireless neural recordings have been delayed by over a year due to the pandemic and associated delays. We have reworked the manuscript throughout to indicate that our facility is equipped for wireless brain recordings.

We do acknowledge that EM shielding is a well-established technique, but we describe innovative modular panels with copper sandwiching that demonstrably reduce EM interference (Figure 1 – supplement 1).

[Editors' note: further revisions were suggested prior to acceptance, as described below.]

Reviewer #1:This revised manuscript is considerably improved in terms of its focus, rigour, and clarity. The authors undertook an immense amount of work through the revision process already, including training another naïve monkey and much more depth and detail included regarding the environment, training, and training outcomes.I have included several recommendations for the authors. I feel strongly that these recommendations and suggestions should be addressed, but I don't think any of these points should preclude publication, and I'd trust the authors discretion in dealing with these suggestions.1) In a few locations the authors still state that naïve monkeys learned the complex cognitive task by socially observing other trained monkeys. In my view, this is an over-interpretation that detracts from the manuscript, which I otherwise find to very interesting.All of the following points are very clear in the revised manuscript: Monkey M2 learned the same-different task faster than monkeys M1 and M3. Monkeys M2 and M4 learned the basic task structure and how to interact with the touchscreen and lick spout while other monkeys were present. Monkey M2 learned the complex cognitive task while alone in the behavioural testing room. Monkey M4 did not learn the complex same-different task, but an alternate strategy to get enough juice to satisfy them for the session. Neither monkey M2 nor M4 learned the same-different task by observing M1 or M3; learning beyond the basic task structure and interaction with the screen happened when the monkeys were alone. These are all very clearly described in the results.Therefore, statements like line 387 ("The above results show that a naïve monkey can learn a complex cognitive task by observing trained monkeys doing the task") are unfounded. The merits of the paper do not rely on monkeys learning a "complex cognitive task" through social learning, so I see no reason to include an over-interpretation results that are clearly explained in the text. The paragraphs starting on Line 445 and Line 509 are clear and measured. These are more accurate than the sentences on lines 397, 465, and the entire paragraph starting on line 501. I suggest that those parts of the text be amended.

Thank you, we have now revised the text throughout to reflect a more measured conclusion regarding the social training. Specifically, we now say that naïve monkeys learned the complex task through a combination of socially observing trained monkeys and solo trial-and-error learning

2) In two locations in the revised manuscript, it is mentioned that the monkeys sometimes were not trained because they did not enter the behavioural testing area voluntarily (line 335, line 1329). It would be of interest for those that might want to replicate this type of facility to know the percentage of training days were missed for each monkey because they did not voluntarily enter the training room.

Thank you for highlighting this, it is indeed important information. In practice there were very few sessions (~5%) in which the monkeys did not come voluntarily for behavioural testing. We have now included this information as well as training details in the Methods.

3) Line 70: This sentence suggests that the methods explain how experimenters isolated individual monkeys, but I didn't see any information about that beyond starting that a positive reinforcement training protocol was used. Given that some days monkeys did not come into the behavioural training room, I infer that there is not an easy way of isolating a specific animal from the group housing, particularly those animals of lower social rank. Whether there was a way to reliably get one specific monkey into the behavioural testing facility is not clear (particularly those of lower rank, who would defer to the higher ranked monkeys when treats are offered).

Thank you for this suggestion. We now describe these details in a separate section of Methods called “Animal training”.

4) Readers might look at e.g. Figure 3, and interpret this to be the upper bound of eye tracking accuracy and gaze reconstruction possible in the testing environment described in the current manuscript. It might be of benefit to the authors and to the readers if the text explicitly stated that further optimizations could be performed but were not because they were not necessary for the present experiments. Given that this is a technical paper, these technical considerations could be discussed for the benefit of others who might be interested in adopting these techniques.

Thank you, we now acknowledge these points in the Methods (line 880-882).

One important example: on line 811, the authors state that calibration was done using a linear mapping. Though this does not at all detract from the merits of this paper, it should be noted that this is not ideal for mapping raw eye data values to eye location. Biquadratic equations provide better estimates of gaze positions. Many papers have been published to this effect, but see https://www.ncbi.nlm.nih.gov/pmc/articles/PMC6721362/ for one example. Also, MonkeyLogic and the Eyelink system rely on the same biquadratic equation for their calibration routines; I'm uncertain about ISCAN.

Thank you, we now acknowledge these points in the Methods (line 880-882).

5) Line 390: The authors state that M4 is dominant over M3, but the dynamic described below is clearly more complicated than that. This sentence initially confused me, and might confuse others.

Thank you, you are absolutely right. We now acknowledge that this dominance reversed at times across sessions.

6) Line 1196: There is no Figure S6. Please reference the correct figure number.

Fixed.

7) Line 1255: "M2" should actually be M3. M2 did not undergo TAT.

Fixed.

Reviewer #2:The manuscript by Jacob et al. describes "A naturalistic environment to study social behaviors and cognitive tasks in freely moving monkeys". There is not much new to the individual components of this environment (more on this below), but this is the first time these components have been put together in this specific way. The main question in this review stage then is whether the manuscript presents a sufficient advance to warrant publication in eLife. The answer would be a clear yes for this reviewer if the new system would allow for fully automatic training inside the monkeys' living environment. This would constitute a major synergy between known components to a point where this combination would be a new system and something that many non-human primate labs would want to adapt. Alas, while there are elements towards such a system presented, this main advance has not been made yet. Therefore, the manuscript, before and after revision, appears fragmented and somewhat unfinished. Still, the description of a step in this direction might be of interest for readers of eLife. Furthermore, eLife establishes criteria for technology papers quite clearly:"… authors will report substantial improvements and extensions of existing technologies. In those cases, the new method must be thoroughly compared and benchmarked against existing methods used in the field. Minor improvements on existing methodologies are unlikely to fare well in review." The current work falls into this category, and thus the onus is on the authors to really demonstrate this advance relative to the past literature. It is here where the manuscript, even after revision, falls short. While touting their own work as a "paradigm shift" or "exciting development", relevant past literature is glossed over and often not even discussed.

Thank you for clarifying your concerns. We do strongly believe that our naturalistic environment fulfils the *eLife* criteria for a substantial improvement and extension of existing technologies. We have used a number of custom-designed components together with existing technologies, all of which have to work together to enable studying complex tasks with high-fidelity gaze tracking in unrestrained monkeys. We have now modified the Introduction to clarify the novelty and technical advances of our study.

A case in point is the treatment of past work on eye tracking in macaque monkeys. The authors state that "there are relatively few studies showing this on macaque monkeys" and cite one reference, Hopper (2021). It might thus appear to a reader not particularly familiar with the subject that this is one example of these few studies, when really Hopper is a review article on "The application of noninvasive, restraint-free eye-tracking methodsfor use with nonhuman primates". Even a quick look into the article shows a great many studies cited and discussed in there. There is no discussion of any of the original work in this review, and there is no mentioning of the subject in the Discussion. This treatment of past literature is easily misleading and scientifically problematic. The authors need to take the past literature seriously and accurately represent the advance that they are making.

We did not mean to trivialize the literature on non-invasive eye tracking studies, since even in our own experience this is a highly non-trivial technical problem. We now acknowledge the fact that the Hopper et al. study is a review, and also cite several original studies related to macaque eye tracking.

On the subject of eye tracking, the revision is moving in the right direction presenting and quantifying some eye tracking data. The authors describe their method as unrestrained. That is debatable, given that a chinrest is used, which would not qualify for unrestrained in human eye tracking for example. The interesting point here is that their subjects are moving to that chin rest by themselves, and one would wish that this advance would be highlighted. Quantification appears to be for a single session of a single subject only. This is not convincing. Even in a badly working system, one could find some good data. A quantification over more than one subject and multiple consecutive sessions would be necessary.

We have now reworked the Introduction to clarify the novelty of our advance in achieving gaze tracking in unrestrained animals. We now report eye tracking data across multiple sessions in all animals and across both same-different (Figure 3, Figure 3 – Supplement 1) and fixation tasks (Figure 3 – Supplement 2).

My third criticism is that claims of observational learning are overstatements of anecdotal observations. It would be better for the scientific quality of the paper, if it mentioned these observations as such, e.g. in supplemental information, focus on the enabling character of the technology for social learning, but not place much emphasis on these few observations (and avoid anthropomorphic interpretations).

We would like to draw your attention to the fact that it is highly time-consuming and labor-intensive to train macaque monkeys on complex tasks such as those reported in this study. As a result, training a larger number of animals is unreasonable and out of scope for the present study. We do realize that field studies often use larger numbers of animals but the tasks are correspondingly much simpler.

We think our basic observation that naïve animals can learn complex tasks through a combination of social observation of trained animals and through solo trial-and-error learning, has been replicated in two monkeys which confirms the utility of this approach for training larger groups of monkeys. We have now carefully reworked our manuscript to avoid anthropomorphizing, separate our key observations from any broader claims, and acknowledge limitations of any broader claims we are making. We have also moved the descriptive analysis into a separate Appendix (Appendix 2).

Reviewer #3:The manuscript describes a new "naturalistic" experimental environment for training and testing macaque monkeys on a popular cognitive task (delayed match-to-sample, also known as a "same/different" task). The manuscript demonstrates that: (1) the animals' eye movements can be monitored with sufficient precision in this environment; (2) the animals can be trained to perform this task with minimal human involvement; (3) the animals can learn faster by watching each other perform the task compared to being trained individually. The authors conclude that this new environment provides a promising platform for studying cognitive and social behaviors, potentially in conjunction with wireless neurophysiological recordings in the future.The manuscript represents an important technical advance in that it demonstrates the feasibility of obtaining robust eye tracking data from monkeys housed in a naturalistic environment. The revised manuscript is an improvement on the initial submission, and likely of interest to researchers who wish to study monkey behavior in a richer and more dynamic setting compared to the traditional lab environment. Given this potential impact, the manuscript is a good addition to the field. Nonetheless, the manuscript can be further improved, as outlined below. Importantly, the authors might wish to re-consider their emphasis on the utility of this new environment for studying (1) social behavior and (2) monkey neurophysiology given that they have not pursued either of these directions fully.

We are glad to note that you found our study interesting and insightful, and thank you for your suggestions. We have reworked our descriptions of the social behaviors and neural activity to qualify our claims.

Suggestions for improvement1) The manuscript would benefit from the removal of the quote at the beginning.

Since several reviewers have suggested it, we have removed this quote.

2) The manuscript would be clearer if the authors refrain from using the word "hybrid". The authors use this word to mean "naturalistic" (as per the manuscript title), and the work "naturalistic" can be used throughout for improved clarity.

Thank you for this suggestion. We have replaced the word “hybrid” with “naturalistic” throughout the main text.

3) It seems inaccurate for the authors to emphasize the study of "social behaviors" in the manuscript title, especially since the data shown area all related to the cognitive "same/different" task. It would be more appropriate to highlight the future utility of using their naturalistic environment for studying social behaviors in the Discussion section.

Thanks for the comment. We have changed the title to “A naturalistic environment to study cognitive tasks in freely moving monkeys.”

4) Similarly, it seems inaccurate for the authors to emphasize the direct utility of their new environment for conducting wireless neurophysiological recordings because they have not yet demonstrated the feasibility of this approach. It would be more appropriate to highlight this point as a future direction in the Discussion section. In particular, these lines should be revised because retaining them would be misleading to the reader:– Line 42: Here we designed a hybrid naturalistic environment with a touchscreen workstation that can be used to record brain activity…–- Line 50: We designed a novel naturalistic environment for recording brain activity…– Line 460: Here, we designed a novel hybrid naturalistic environment with a touchscreen workstation that can be used to record brain activity– Line 560: In sum, our environment represents an important first step in turning the traditional monkey neurophysiology paradigm on its head….

Thank you for your suggestions. We have now highlighting the prospect of recording neural activity as a future direction in the Discussion, and have acknowledged the utility of many design elements as useful for future neural recordings.

(5) The Discussion section would benefit tremendously from an additional paragraph about the many remaining challenges/limitations to neurophysiological recordings in this naturalistic environment. Otherwise, the reader might walk away thinking that achieving this next step of neural recordings is trivial when it is not.

Thank you, we agree with you that achieving neural recordings has its own challenges which we now acknowledge in the Discussion in a separate section.